# WASSERSTEIN BARYCENTER MODEL ENSEMBLING

**Pierre Dognin**[*], **Igor Melnyk**[*] ,**Youssef Mroueh**[*] , **Jerret Ross**[*], **Cicero Dos Santos**[*] & **Tom Sercu**[*]
IBM Research & MIT-IBM Watson AI Lab
[*] Alphabetical order; Equal contribution
{pdognin,mroueh,rossja,cicerons}@us.ibm.com,
{igor.melnyk,tom.sercu1}@ibm.com

## ABSTRACT

In this paper we propose to perform model ensembling in a multiclass or a multilabel learning setting using Wasserstein (W.) barycenters. Optimal transport metrics, such as the Wasserstein distance, allow incorporating semantic side information such as word embeddings. Using W. barycenters to find the consensus between models allows us to balance confidence and semantics in finding the agreement between the models. We show applications of Wasserstein ensembling in attribute-based classification, multilabel learning and image captioning generation. These results show that the W. ensembling is a viable alternative to the basic geometric or arithmetic mean ensembling.

## 1 INTRODUCTION

Model ensembling consists in combining many models into a stronger, more robust and more accurate model. Ensembling is ubiquitous in machine learning and yields improved accuracies across multiple prediction tasks such as multi-class or multi-label classification. For instance in deep learning, output layers of Deep Neural Networks(DNNs), such as softmaxes or sigmoids, are usually combined using a simple arithmetic or geometric mean. The arithmetic mean rewards confidence of the models while the geometric means seeks the consensus across models.

What is missing in the current approaches to models ensembling, is the ability to incorporate side information such as class relationships represented by a graph or via an embedding space. For example a semantic class can be represented with a finite dimensional vector in a pretrained word embedding space such as GloVe (Pennington et al., 2014). The models' predictions can be seen as defining a distribution in this label space defined by word embeddings: if we denote $p_i$ to be the confidence of a model on a bin corresponding to a word having an embedding $x_i$, the distribution on the label space is therefore $p = \sum_i p_i \delta_{x_i}$. In order to find the consensus between many models predictions, we propose to achieve this consensus within this representation in the label space. In contrast to arithmetic and geometric averaging, which are limited to the independent bins' confidence, this has the advantage of carrying the semantics to model averaging via the word embeddings. More generally this semantic information can be encoded via cost a matrix $C$, where $C_{ij}$ encodes the dissimilarity between semantic classes $i$ and $j$, and $C$ defines a ground metric on the label space.

To achieve this goal, we propose to combine model predictions via Wasserstein (W.) barycenters (Agueh & Carlier, 2011), which enables us to balance the confidence of the models and the semantic side information in finding a consensus between the models. Wasserstein distances are a naturally good fit for such a task, since they are defined with respect to a ground metric in the label space of the models, which carry such semantic information. Moreover they enable the possiblity of ensembling predictions defined on different label sets, since the Wasserstein distance allows to align and compare those different predictions. Since their introduction in (Agueh & Carlier, 2011) W. barycenter computations were facilitated by entropic regularization (Cuturi, 2013) and iterative algorithms that rely on iterative Bregman projections (Benamou et al., 2015). Many applications have used W. barycenters in Natural Language Processing (NLP), clustering and graphics. We show in this paper that W. barycenters are effective in model ensembling and in finding a semantic consensus, and can be applied to a wide range of problems in machine learning (Table 1).

The paper is organized as follows: In Section 2 we revisit geometric and arithmetic means from a geometric viewpoint, showing that they are $\ell_2$ and Kullback Leibler divergence KL (extended

KL divergence) barycenters respectively. We give a brief overview of optimal transport metric and W. barycenters in Section 3. We highlight the advantages of W. barycenter ensembling in terms of semantic smoothness and diversity in Section 4. Related work on W. barycenters in Machine learning are presented in Section 5. Finally we show applications of Wasserstein ensembling on attribute based classification, multi-label learning and image captioning in Section 6.

## 2 WASSERSTEIN BARYCENTERS FOR MODEL ENSEMBLING

**Normalized and Unnormalized predictions Ensembling.** In deep learning, predictions on a label space of fixed size $M$ are usually in one of two forms: a) *normalized probabilities*: in a multi-class setting, the neural network outputs a probability vector (normalized through softmax), where each bin corresponds to a semantic class; b) *unnormalized positive scores*: in a multi-label setting, the outputs of $M$ independent logistic units are unnormalized positive scores, where each unit corresponds to the presence or the absence of a semantic class.

Model ensembling in those two scenarios has long history in deep learning and more generally in machine learning (Breiman, 1996; Freund & Schapire, 1999; Wolpert, 1992) as they lead to more robust and accurate models. As discussed in the introduction, two methods have been prominent in model ensembling due to their simplicity: majority vote using the arithmetic mean of predictions, or consensus based using the geometric mean.

**Revisiting Arithmetic and Geometric Means from a geometric viewpoint.** Given $m$ predictions $\mu_\ell$, and weights $\lambda_\ell \geq 0$ such that $\sum_{\ell=1}^{m} \lambda_\ell = 1$, the weighted arithmetic mean is given by $\bar{\mu}_a = \sum_{\ell=1}^{m} \lambda_\ell \mu_\ell$, and the weighted geometric mean by $\bar{\mu}_g = \Pi_{\ell=1}^{m}(\mu_\ell^{\lambda_\ell})$.

It is instructive to reinterpret the arithmetic and geometric mean as weighted Frechet means (Definition 1) (Zemel & Panaretos, 2017).

**Definition 1.** *[Weighted Frechet Mean] Given a distance $d$ and $\{(\lambda_\ell, \mu_\ell), \lambda_\ell > 0, \mu_\ell \in \mathbb{R}_{+}^{M}\}_{\ell=1...m}$, the Frechet mean is defined as follows: $\bar{\mu} = \arg\min_\rho \sum_{\ell=1}^{m} \lambda_\ell d(\rho, \mu_\ell)$.*

It is easy to prove (Appendix F) that the arithmetic mean corresponds to a Frechet mean for $d = \|.\|_2^2$ (the $\ell_2$ Euclidean distance). A less known fact is that the geometric mean corresponds to a Frechet Mean for $d = \widetilde{\text{KL}}$, where $\widetilde{\text{KL}}$ is the extended KL divergence to unnormalized measures: $\widetilde{\text{KL}}(p, q) = \sum_i p_i \log\left(\frac{p_i}{q_i}\right) - p_i + q_i$. We give proofs and properties of arithmetic and geometric mean in Appendix F.

Following this geometric viewpoint, in order to incorporate the semantics of the target space in model ensembling, we need to use a distance $d$ that takes advantage of the underlying geometry of the label space via a cost matrix $C$ when comparing positive measures. Optimal transport (OT) metrics such as Wasserstein-2 have this property since they are built on an explicit cost matrix defining pairwise distance between the semantic classes. In this paper we propose to use the Frechet means with Wasserstein distance ($d = W_2^2$) for model ensembling, i.e. use Wasserstein barycenters (Agueh & Carlier, 2011) for model ensembling:

$$\bar{\mu}_w = \arg\min_\rho \sum_{\ell=1}^{m} \lambda_\ell W_2^2(\rho, \mu_\ell).$$

Intuitively, the barycenter looks for a distribution $\rho$ (a histogram ) that is close to all the base distributions $\mu_\ell$ in the Wasserstein sense. In our context transporting the consensus $\rho$ to each individual model $\mu_\ell$ should have a minimal cost, where the cost is defined by the distance in the word embedding space.

## 3 WASSERSTEIN BARYCENTERS

Wasserstein distances were originally defined between normalized probability vectors (Balanced OT) (Villani, 2008; Peyré & Cuturi, 2017), but they have been extended to deal with unnormalized measures and this problem is referred to as unbalanced OT (Chizat et al., 2018; Frogner et al., 2015). Motivated by the multi-class and the multi-label ensembling applications, in the following we present a brief overview of W. barycenters in the balanced and unbalanced cases.

### 3.1 Optimal Transport Metrics

**Balanced OT.** Given $p \in \Delta_N$, where $\Delta_N = \{p \in \mathbb{R}^N, p_k \geq 0, \sum_{k=1}^N p_k = 1\}$, $p$ represents histograms on source label space $\Omega^S = \{x_i \in \mathbb{R}^d, i = 1 \dots N\}$, for e.g words embeddings. Consider similarly $q \in \Delta_M$ representing histograms whose bins are defined on a target label space $\Omega^T = \{y_j \in \mathbb{R}^d, j = 1 \dots M\}$. Consider a cost function $c(x, y)$, (for example $c(x, y) = \|x - y\|^2$). Let $C$ be the matrix in $\in \mathbb{R}^{N \times M}$ such that $C_{ij} = c(x_i, y_j)$. $1_N$ denotes a vector with all ones. Let $\gamma \in \mathbb{R}^{N \times M}$ be a coupling matrix whose marginals are $p$ and $q$ such that: $\gamma \in \Pi(p, q) = \{\gamma \in \mathbb{R}^{N \times M}, \gamma 1_M = p, \gamma^\top 1_N = q\}$. The optimal transport metric is defined as follows:

$$W(p, q) = \min_{\gamma \in \Pi(p,q)} \{\langle C, \gamma \rangle = \sum_{ij} C_{ij} \gamma_{ij}.\} \tag{1}$$

When $c(x, y) = \|x - y\|_2^2$, this distance corresponds to the so called Wasserstein$-2$ distance $W_2^2$.

**Unbalanced OT.** When $p$ and $q$ are unnormalized and have different total masses, optimal transport metrics have been extended to deal with this unbalanced case. The main idea is in relaxing the set $\Pi(p, q)$ using a divergence such as the extended KL divergence: $\widetilde{KL}$. (Chizat et al., 2018) define for $\lambda > 0$ the following generalized Wasserstein distance between unnormalized measures:

$$W_{\text{unb}}(p, q) = \min_\gamma \langle C, \gamma \rangle + \lambda \widetilde{KL}(\gamma 1_M, p) + \lambda \widetilde{KL}(\gamma^\top 1_N, q). \tag{2}$$

### 3.2 Balanced and Unbalanced Wasserstein in Models Ensembling

Throughout the paper we consider $m$ discrete prediction vectors $\mu_\ell \in \mathbb{R}_+^{N_\ell}, \ell = 1 \dots m$ defined on a discrete space (word embeddings for instance) $\Omega_\ell^S = \{x_i^\ell \in \mathbb{R}^d, i = 1 \dots N_\ell\}$. We refer to $\Omega_\ell^S$ as *source* spaces. Our goal is to find a consensus prediction $\bar{\mu}_w \in \mathbb{R}_+^M$ defined on a discrete *target* space $\Omega^T = \{y_j \in \mathbb{R}^d, j = 1 \dots M\}$. Let $C_\ell \in \mathbb{R}^{N_\ell \times M}$ be the cost matrices, $C_{\ell,i,j} = c(x_i^\ell, y_j)$.

**Balanced W. Barycenters: Normalized predictions.** The W. barycenter (Agueh & Carlier, 2011) of normalized predictions is defined as follows: $\bar{\mu}_w = \arg\min_\rho \sum_{\ell=1}^m \lambda_\ell W(\rho, \mu_\ell)$, for the Wasserstein distance $W$ defined in equation (1). Hence one needs to solve the following problem, for $m$ coupling matrices $\gamma_\ell, \ell = 1 \dots m$:

$$\min_\rho \min_{\gamma_\ell \in \Pi(\mu_\ell, \rho), \ell=1 \dots m} \sum_{\ell=1}^m \lambda_\ell \langle C_\ell, \gamma_\ell \rangle. \tag{3}$$

**Unbalanced W. Barycenters: Unnormalized predictions.** Similarly the W. barycenter of unnormalized predictions is defined as follows: $\bar{\mu}_w = \arg\min_\rho \sum_{\ell=1}^m \lambda_\ell W_{\text{unb}}(\rho, \mu_\ell)$, for the Generalized Wasserstein distance $W_{\text{unb}}$ defined in equation (2). Hence the unbalanced W. barycenter problem (Chizat et al., 2018) amounts to solving , for $m$ coupling matrices $\gamma_\ell, \ell = 1 \dots m$:

$$\min_\rho \min_{\gamma_\ell, \ell=1 \dots m} \sum_{\ell=1}^m \lambda_\ell \left( \langle C_\ell, \gamma_\ell \rangle + \lambda \widetilde{KL}(\gamma_\ell 1_M, \mu_\ell) + \lambda \widetilde{KL}(\gamma_\ell^\top 1_{N_\ell}, \rho) \right). \tag{4}$$

### 3.3 Computation via Entropic Regularization and Practical Advantages

**Entropic Regularized Wasserstein Barycenters Algorithms.** The computation of the Wasserstein distance grows super-cubically in the number of points. This issue was alleviated by the introduction of the entropic regularization (Cuturi, 2013) to the optimization problem making it strongly convex. Its solution can be found using scaling algorithms such as the so called Sinkhorn algorithm. For any positive matrix $\gamma$, the entropy is defined as follows: $H(\gamma) = -\sum_{ij} \gamma_{ij}(\log(\gamma_{ij}) - 1)$. The entropic regularized OT distances in the balanced and unbalanced case become, for a hyperparameter $\varepsilon > 0$:

$$W_\varepsilon(p, q) = \min_{\gamma \in \Pi(p,q)} \langle C, \gamma \rangle - \varepsilon H(\gamma),$$

$$W_{\text{unb},\varepsilon}(p, q) = \min_\gamma \langle C, \gamma \rangle + \lambda \widetilde{KL}(\gamma 1_M, p) + \lambda \widetilde{KL}(\gamma^\top 1_N, q) - \varepsilon H(\gamma)$$

for $\varepsilon \to 0$, $W_\varepsilon$ and $W_{\text{unb},\varepsilon}$ converge to the original OT distance, and for higher value of $\varepsilon$ we obtain the so called Sinkhorn divergence that allows for more diffuse transport between $p$ and $q$. Balanced and unbalanced W. barycenters can be naturally defined with the entropic regularized OT distance as follows: $\min_\rho \sum_{\ell=1}^m \lambda_\ell W_\varepsilon(\rho, \mu_\ell)$ and $\min_\rho \sum_{\ell=1}^m \lambda_\ell W_{\text{unb},\varepsilon}(\rho, \mu_\ell)$ respectively. This regularization leads to simple iterative algorithms (Benamou et al., 2015; Chizat et al., 2018) (for more details we refer the interested reader to (Chizat et al., 2018) and references therein) for computing W. barycenters that are given in Algorithms 1 and 2.

| **Algorithm 1:** Balanced Barycenter for Multi-class Ensembling (Benamou et al., 2015) | **Algorithm 2:** Unbalanced Barycenter for Multi-label Ensembling (Chizat et al., 2018) |
|---|---|
| **Inputs:** $\varepsilon$, $C_\ell$ ($|\text{source}| \times |\text{target}|$), $\lambda_\ell$, $\mu_\ell$ 
 **Initialize** 
 $K_\ell = \exp(-C_\ell/\varepsilon)$, $v_\ell \leftarrow 1_M, \forall \ell = 1 \ldots m$ 
 **for** $i = 1 \ldots$ Maxiter **do** 
 $\quad u_\ell \leftarrow \dfrac{\mu_\ell}{K_\ell v_\ell}, \forall \ell = 1 \ldots m$ 
 $\quad p \leftarrow \exp\left(\sum_{\ell=1}^m \lambda_\ell \log\left(K_\ell^\top u_\ell\right)\right) = \Pi_{\ell=1}^m (K_\ell^\top u_\ell)^{\lambda_\ell}$ 
 $\quad v_\ell \leftarrow \dfrac{p}{K_\ell^\top u_\ell} \ell = 1 \ldots m$ 
 **end for** 
 **Output:** $p$ | **Inputs:** $\varepsilon$, $C_\ell$ ($|\text{source}| \times |\text{target}|$), $\lambda_\ell$, $\lambda$, $\mu_\ell$ 
 **Initialize** 
 $K_\ell = \exp(-C_\ell/\varepsilon)$, $v_\ell \leftarrow 1, \forall \ell = 1 \ldots m$ 
 **for** $i = 1 \ldots$ Maxiter **do** 
 $\quad u_\ell \leftarrow \left(\dfrac{\mu_\ell}{K_\ell v_\ell}\right)^{\frac{\lambda}{\lambda+\varepsilon}}, \forall \ell = 1 \ldots m$ 
 $\quad p \leftarrow \left(\sum_{\ell=1}^m \lambda_\ell \left(K_\ell^\top u_\ell\right)^{\frac{\varepsilon}{\lambda+\varepsilon}}\right)^{\frac{\lambda+\varepsilon}{\varepsilon}},$ 
 $\quad v_\ell \leftarrow \left(\dfrac{p}{K_\ell^\top u_\ell}\right)^{\frac{\lambda}{\lambda+\varepsilon}} \ell = 1 \ldots m$ 
 **end for** 
 **Output:** $p$ |

We see that the output of Algorithm 1 is the geometric mean of $K_\ell^\top u_\ell, \ell = 1 \ldots m$, where $K_\ell$ is a Gaussian kernel with bandwidth $\varepsilon$ the entropic regularization parameter. Note $v_\ell^*, \ell = 1 \ldots m$ the values of $v_\ell$ at convergence of Algorithm 1. The entropic regularized W. barycenter can be written as follows: $\exp\left(\sum_{\ell=1}^M \lambda_\ell \left(\log(K_\ell^\top \frac{\mu_\ell}{K_\ell v_\ell^*})\right)\right)$. We see from this that $K_\ell$ appears as matrix product multiplying individual models probability $\mu_\ell$ and the quantities $v_\ell^*$ related to Lagrange multipliers. This matrix vector product with $K_\ell$ ensures probability mass transfer between semantically related classes i.e between items that has entries $K_{\ell,ij}$ with high values.

**Remark 1** (The case $K_\ell = K = I$)**.** *As the kernel $K$ in Algorithm 1 approaches $I$ (identity) (this happens when $\varepsilon \to 0$), the alternating Bregman projection of (Benamou et al., 2015) for balanced W. barycenter converges to the geometric mean $\bar\mu_g = \Pi_{\ell=1}^m (\mu_\ell)^{\lambda_\ell}$. We prove this in Appendix D. When $K = I$ the fixed point of Algorithm 1 reduces to geometric mean, and hence diverges from the W. barycenter. Note that $K$ approaches identity as $\varepsilon \to 0$, and in this case we don't exploit any semantics.*

**Wasserstein Ensembling in Practice.** Table 1 gives a summary of machine learning tasks that can benefit from Wasserstein Ensembling, and highlights the source and target domains as well as the corresponding kernel matrix $K$. In the simplest case $\Omega_\ell^S = \Omega^T$ and $N_\ell = M$ for all $\ell$, this corresponds to the case we discussed in multi-class and multi-labels ensemble learning, W. barycenters allows to balance semantics and confidence in finding the consensus. The case where source and target spaces are different is also of interest, we give here an application example in attribute based classification : $\mu_\ell$ corresponds to prediction on a set of attributes and we wish to make predictions through the W. barycenter on a set of labels defined with those attributes. See Section 6.1. Table 12 in Appendix C gives other possible machine learning applications beyond the one explored in this paper. Appendix E discusses the computational complexity of Alg. 1 and 2.

# 4 THEORETICAL ADVANTAGES OF WASSERSTEIN BARYCENTERS IN MODELS ENSEMBLING

**Smoothness of the Wasserstein Barycenter within Semantically Coherent Clusters.** In this section we consider $\Omega_S^\ell = \Omega_T = \Omega$, i.e the W. barycenters and all individual models are defined on the same label space. When we are ensembling models, one desiderata is to have an *accurate* aggregate model. *Smoothness and Diversity* of the predictions of the ensemble is another desiderata as we often want to supply many diverse hypotheses. In the context of sequence generation in language modeling such as image captioning, machine translation or dialog, this is very important

| | Source Domains (models) | Target Domain (Barycenter) | Kernel $K$ (OT cost matrix) | Arithmetic Geometric apply |
|---|---|---|---|---|
| Multi-class Learning (Balanced OT) | $\mu_\ell \in \Delta_N, \ell = 1 \dots m$ Histograms of size $N$ | $p \in \Delta_N$ Histogram of size N | $K_{ij} = e^{-\frac{\|x_i - x_j\|^2}{\varepsilon}}$ $x_i$ word embedding of category $i$ (See Section 6.3 for e.g GloVe or Visual w2v) | ✓ |
| Multi-label Learning (Unbalanced OT) | $\mu_\ell \in [0,1]^N, \ell = 1 \dots m$ Soft multi-labels | $p \in [0,1]^N$ Soft multi-label | $K_{ij}$ adjacency weight in a knowledge graph $K_{ij}$= co-occurrence of item $i$ and $j$ (See Section 6.2 for other choices) | ✓ |
| Cost-sensitive Classification (Balanced OT) | $\mu_\ell \in \Delta_N, \ell = 1 \dots m$ Histograms of size $N$ | $p \in \Delta_N$ Histogram of size $N$ | $K_{ij}$ User ratings of similarities user-defined costs for confusion of $i$ and $j$ e.g: binary matrix for synonyms (Section 6.3) | ✓ |
| Attribute to Categories (Unbalanced OT) | $\mu_\ell \in [0,1]^N, \ell = 1 \dots m$ Soft multi-labels: $N$ attributes | $p \in \Delta_M$ Histogram of size $M$ $M$ categories | $K_{ij}$ presence or absence of attribute $i$ in class $j$ (See Section 6.1) | ✗ |

Table 1: Machine learning tasks where W. Ensembling can be applied. Note that W. barycenter allows ensembling different source domains to another target domain as in attributes to category.

as we use beam search on the predictions, diversity and smoothness of the predictions become key to the creativity and the composition of the sequence generator in order to go beyond "baby talk" and vanilla language based on high count words in the training set. Hence we need to increase the entropy of the prediction by finding a *semantic consensus* whose predictions are *diverse* and *smooth* on semantically coherent concepts without compromising accuracy. We will show in the following proposition that the W. barycenter allows such aggregation:

**Proposition 1** (Properties of Wasserstein Barycenters). *Let $\nu$ be the target distribution (an oracle) defined on a discrete space $\Omega = \{x_1, \dots x_K, x_j \in \mathbb{R}^d\}$ (word embedding space) and $\mu_\ell, \ell = 1 \dots m$ be $m$ estimates of $\nu$. Assume $W_2^2(\mu_\ell, \nu) \leq \varepsilon_\ell$. The W. barycenter $\bar{\mu}_w$ of $\{\mu_\ell\}$ satisfies the following:*
*1) Semantic Accuracy (Distance to an oracle). We have: $W_2^2(\bar{\mu}_w, \nu) \leq 4 \sum_{\ell=1}^m \lambda_\ell W_2^2(\mu_\ell, \nu)$. Assume that $W_2^2(\mu_\ell, \nu) \leq \varepsilon_\ell$, then we have: $W_2^2(\bar{\mu}_w, \nu) \leq 4 \sum_{\ell=1}^m \lambda_\ell \varepsilon_\ell$.*
*2) Diversity. The diversity of the W. barycenter depends on the diversity of the models with respect to the Wasserstein distance (pairwise Wasserstein distance between models): $W_2^2(\bar{\mu}_w, \mu_k) \leq \sum_{\ell \neq k}^m \lambda_\ell W_2^2(\mu_\ell, \mu_k), \ \forall k = 1, \dots m.$*
*3) Smoothness in the embedding space. Define the smoothness energy $\mathscr{E}(\rho) = \sum_{ij} \|x_i - x_j\|^2 \rho_i \rho_j$. We have: $\mathscr{E}(\bar{\mu}_w) \leq \sum_{\ell=1}^m \lambda_\ell \mathscr{E}(\mu_\ell)$. The W. barycenter is smoother in the embedding space than the individual models.*
*4) Entropy . Let $H(\rho) = -\sum_{i=1}^K \rho_i \log(\rho_i)$, we have: $H(\bar{\mu}_w) \geq \sum_{\ell=1}^m \lambda_\ell H(\mu_\ell)$.*

*Proof.* The proof is given in Appendix F. □

We see from Proposition 1 that the W. barycenter preserves accuracy, but has a higher entropy than the individual models. This entropy increase is due to an improved smoothness on the embedding space: words that have similar semantics will have similar probability mass assigned in the barycenter. The diversity of the barycenter depends on the Wasserstein pairwise distance between the models: the W. barycenter output will be less diverse if the models have similar semantics as measured by the Wasserstein distance. The proof of proposition 1 relies on the notion of convexity along generalized geodesics of the Wasserstein 2 distance (Agueh & Carlier, 2011). Propositions 2 and 3 in Appendix F give similar results for geometric and arithmetic mean, note that the main difference is that the guarantees are given in terms of $\widetilde{KL}$ and $\ell_2$ respectively, instead of $W_2$.

In order to illustrate the diversity and smoothness of the W. barycenter, we give here a few examples of the W. barycenter on a vocabulary of size 10000 words, where the cost matrix is constructed from word synonyms ratings, defined using Power Thesaurus or using GloVe word embeddings (Pennington et al., 2014). We compute the W. barycenter (using Algorithm 1) between softmax outputs of 4 image captioners trained with different random seeds and objective functions. Figure 4 shows the W. barycenter as well as the arithmetic and geometric mean. It can be seen that the W. barycenter has higher entropy and is smooth along semantics (synonyms or semantics in the GloVe space) and hence more diverse than individual models. Table 2 shows top 15 words of barycenter, arithmetic and geometric means, from which we see that indeed the W. barycenter outputs clusters according to semantics. In order to map back the words $x_j$ that have high probability in the W. barycenter to an individual model $\ell$, we can use the couplings $\gamma^\ell$ as follows: $\gamma_{ij}^\ell$ is the coupling between word $j$ in the barycenter and word $i$ in model $\ell$. Examples are given in supplement in Table 9.

| Rank | W. Barycenter | | Arithmetic | | Geometric | | Model 1 | | Model 2 | | Model 3 | | Model 4 | |
|---|---|---|---|---|---|---|---|---|---|---|---|---|---|---|
| 0 | car | 03.73 | car | 45.11 | car | 41.94 | car | 61.37 | car | 62.25 | car | 33.25 | car | 46.88 |
| 1 | van | 03.50 | fashion | 04.37 | truck | 02.23 | cars | 02.79 | cars | 03.16 | fashion | 18.15 | truck | 07.74 |
| 2 | truck | 03.49 | truck | 02.92 | black | 01.67 | parking | 02.62 | white | 02.22 | black | 03.08 | bus | 04.78 |
| 3 | vehicle | 03.46 | buildin | 02.10 | train | 01.51 | vehicle | 01.93 | black | 01.95 | truck | 02.29 | vehicle | 03.46 |
| 4 | wagon | 03.32 | bus | 02.00 | fashion | 01.49 | model | 01.75 | train | 01.68 | red | 01.88 | red | 02.20 |
| 5 | automob | 03.32 | black | 01.79 | bus | 01.30 | train | 01.26 | passeng | 01.33 | photo | 01.57 | van | 01.93 |
| 6 | coach | 02.99 | train | 01.73 | vehicle | 01.14 | truck | 01.22 | model | 01.24 | parking | 01.52 | fashion | 01.74 |
| 7 | auto | 02.98 | parking | 01.55 | photo | 01.01 | buildin | 01.17 | photo | 01.21 | city | 01.41 | passeng | 01.56 |
| 8 | bus | 02.85 | vehicle | 01.49 | van | 01.01 | black | 01.04 | truck | 01.15 | train | 01.30 | pickup | 01.37 |
| 9 | sedan | 02.71 | cars | 01.41 | red | 01.01 | van | 01.04 | red | 01.15 | buildin | 00.74 | black | 01.29 |
| 10 | cab | 02.70 | photo | 01.29 | parking | 00.94 | fashion | 00.82 | silver | 01.03 | fashion | 00.72 | train | 00.79 |
| 11 | wheels | 02.70 | red | 01.26 | buildin | 00.88 | suv | 00.69 | vehicle | 00.78 | bus | 00.71 | style | 00.68 |
| 12 | buggy | 02.70 | van | 01.18 | cars | 00.81 | automob | 00.67 | van | 00.75 | style | 00.69 | model | 00.59 |
| 13 | motor | 02.39 | white | 01.04 | passeng | 00.71 | parked | 00.57 | buildin | 00.71 | time | 00.67 | fire | 00.57 |
| 14 | jeep | 02.31 | passeng | 00.92 | white | 00.67 | picture | 00.55 | bus | 00.70 | old | 00.58 | white | 00.52 |

Table 2: Sample output (top 15 words) of W. barycenter (Algorithm 1), arithmetic and geometric means based on four captioner models. Each row shows a word and a corresponding probability over the vocabulary (as a percentage). W. Barycenter has higher entropy, spreading the probability mass over the synonyms and words related to the top word "car" and downweights the irrelevant objects (exploiting the side information $K$). Simple averaging techniques, which use only the confidence information, mimick the original model outputs. Figure 4 in Appendix gives a histogram view.

**Controllable Entropy via Regularization.** As the entropic regularization parameter $\varepsilon$ increases the distance of the kernel $K$ from identity $I$ increases and the entropy of the optimal couplings $\gamma_\ell$, $(H(\gamma_\ell))$ increases as well. Hence the entropy of entropic regularized W. Barycenter is controllable via the entropic regularization parameter $\varepsilon$. In fact since the barycenter can be written as $\bar{\mu}_w = \gamma_\ell^\top 1_{N_\ell}$, one can show that (Lemma 2 in Appendix):

$$H(\bar{\mu}_w) + H(\mu_\ell) \geq -\sum_{i,j} \gamma_{i,j}^\ell \log(\gamma_{ij}^\ell), \forall \ell = 1 \dots m,$$

As epsilon increases the right-hand side of the inequality increases and so does $H(\bar{\mu}_w)$. This is illustrated in Tables 3 and 8, we see that the entropy of the (entropic regularized) W. barycenter increases as the distance of the kernel $K$ to identity increases ($\|K - I\|_F$ increases as $\varepsilon$ increases ) and the output of the W. barycenter remains smooth within semantically coherent clusters.

| Rank | 109.7 | | 79.8 | | 59.4 | | 43.3 | | 15.8 | | 0.25 | |
|---|---|---|---|---|---|---|---|---|---|---|---|---|
| 0 | car | 10.51 | car | 12.79 | car | 15.52 | car | 19.22 | car | 33.60 | car | 41.94 |
| 1 | truck | 10.30 | vehicle | 10.24 | vehicle | 10.48 | vehicle | 10.26 | vehicle | 05.64 | truck | 02.23 |
| 2 | vehicle | 09.73 | truck | 09.16 | auto | 08.87 | auto | 08.42 | auto | 03.45 | black | 01.67 |
| 3 | auto | 08.46 | auto | 08.82 | truck | 07.96 | truck | 06.59 | truck | 03.31 | train | 01.51 |
| 4 | machine | 08.17 | machine | 06.17 | machine | 04.33 | machine | 02.55 | black | 01.67 | fashion | 01.49 |
| 5 | black | 01.67 | black | 01.67 | black | 01.67 | black | 01.67 | bus | 01.54 | bus | 01.30 |
| 6 | fashion | 01.49 | fashion | 01.49 | fashion | 01.49 | fashion | 01.49 | fashion | 01.49 | vehicle | 01.14 |
| 7 | red | 01.06 | red | 01.05 | van | 01.06 | van | 01.12 | van | 01.08 | photo | 01.01 |
| 8 | white | 00.98 | van | 00.99 | red | 01.04 | bus | 01.11 | red | 01.01 | van | 01.01 |
| 9 | parking | 00.94 | parking | 00.94 | parking | 00.94 | red | 01.03 | photo | 00.96 | red | 01.01 |
| 10 | van | 00.91 | white | 00.91 | bus | 00.88 | parking | 00.94 | parking | 00.94 | parking | 00.94 |
| 11 | cars | 00.81 | cars | 00.81 | white | 00.85 | cars | 00.81 | cars | 00.81 | buildin | 00.88 |
| 12 | coach | 00.73 | bus | 00.69 | cars | 00.81 | white | 00.79 | train | 00.81 | cars | 00.81 |
| 13 | photogr | 00.64 | coach | 00.67 | photo | 00.69 | photo | 00.77 | buildin | 00.72 | passeng | 00.71 |
| 14 | photo | 00.57 | photo | 00.63 | coach | 00.61 | coach | 00.55 | white | 00.68 | white | 00.67 |

Table 3: Controllable Entropy of regularized Wasserstein Barycenter (Algorithm 1). Output (top 15 words) for a synonyms-based similarity matrix $K$ under different regularization $\varepsilon$ (which controls the distance of $K$ to identity $I$, $\|K - I\|_F$). As $\varepsilon$ decreases, $\|K - I\|_F$ also decreases, i.e., $K$ approaches identity matrix, and the entropy of the output of Algorithm 1 decreases. Note that the last column, corresponding to very small entropic regularization, coincides with the output from geometric mean in Figure 2 (for $K = I$, the Algorithm 1 outputs geometric mean as a barycenter).

## 5 RELATED WORK

**Wasserstein Barycenters in Machine Learning.** Optimal transport is a relatively new comer to the machine learning community. The entropic regularization introduced in (Cuturi, 2013) fostered many applications and computational developments. Learning with a Wasserstein loss in a multi-label setting was introduced in (Frogner et al., 2015), representation learning via the Wasserstein discriminant analysis followed in (Flamary et al., 2016). More recently a new angle on generative adversarial networks learning with the Wasserstein distance was introduced in (Arjovsky et al., 2017; Genevay et al., 2017; Salimans et al., 2018). Applications in NLP were pioneered by the work on Word Mover Distance (WMD) on word embeddings of (Kusner et al., 2015). Thanks to new algorithmic developments (Cuturi & Doucet, 2014; Benamou et al., 2015) W. barycenters have been applied to various problems : in graphics (Solomon et al., 2015), in clustering (Ye et al., 2017), in dictionary learning (Schmitz et al., 2018), in topic modeling (Xu et al., 2018), in bayesian averaging (Rios et al., 2018), and in learning word and sentences embeddings (Muzellec & Cuturi, 2018; Pal

Singh et al., 2018) etc. Most of these applications of W. barycenter focus on learning balanced barycenters in the embedding space (like learning the means of the clusters in clustering), in our ensembling application we assume the embeddings given to us (such as GloVe word embedding ) and compute the barycenter at the predictions level. Finally incorporating side information such as knowledge graphs or word embeddings in classification is not new and has been exploited in diverse ways at the level of individual model training via graph neural networks (Marino et al., 2017; Deng et al., 2014), in the framework of W. barycenter we use this side information at the ensemble level.

## 6 APPLICATIONS

In this Section we evaluate W. barycenter ensembling in the problems of attribute-based classification, multi-label prediction and in natural language generation in image captioning.

### 6.1 ATTRIBUTE BASED CLASSIFICATION

As a first simple problem we study object classification based on attribute predictions. We use Animals with Attributes (Xian et al., 2017) which has 85 attributes and 50 classes. We have in our experiments 2 attributes classifiers to predict the absence/presence of each of the 85 attributes independently, based on (1) resnet18 and (2) resnet34 (He et al., 2016) input features while training only the linear output layer (following the details in Section 6.2). We split the data randomly in 30322 / 3500 / 3500 images for train / validation / test respectively. We train the attribute classifiers on the train split.

Based on those two attributes detectors we would like to predict the 50 categories using unbalanced W. barycenters using Algorithm 2. Note that in this case the source domain is the set of the 85 attributes and the target domain is the set of 50 animal categories. For Algorithm 2 we use a column-normalized version of the binary animal/attribute matrix as $K$ matrix ($85 \times 50$), such that per animal the attribute indicators sum to 1. We selected the hyperparameters $\varepsilon = 0.3$ and $\lambda = 2$ on the validation split and report here the accuracies on the test split.

| Accuracy | resnet18 alone | resnet34 alone | Arithmetic | Geometric | W. Barycenter |
|---|---|---|---|---|---|
| Validation | 0.7771 | 0.8280 | 0.8129 | 0.8123 | **0.8803** |
| Test | 0.7714 | 0.8171 | 0.8071 | 0.8060 | **0.8680** |

Table 4: Attribute-based classification. The W. barycenter ensembling achieves better accuracy by exploiting the cross-domain similarity matrix $K$, compared to a simple linear-transform of probability mass from one domain to another as for the original models or their simple averages.

As a baseline for comparison, we use arithmetic mean ($\bar{\mu}_a$) and geometric mean ($\bar{\mu}_g$) ensembling of the two attribute classifiers resnet18 and resnet34. Then, using the same matrix $K$ as above, we define the probability of category c (animal) as $p(c|\mu) = K^\top \bar{\mu}$ (for $\bar{\mu} = \bar{\mu}_a$ and $\bar{\mu}_g$ resp.). We see from Table 4 that W. barycenter outperforms arithmetic and geometric mean on this task and shows its potential in attribute based classification.

### 6.2 MULTI-LABEL PREDICTION

For investigating W. barycenters on a multi-label prediction task, we use MS-COCO (Lin et al., 2014) with 80 objects categories. MS-COCO is split into training ($\approx$82K images), test ($\approx$35K), and validation (5K) sets, following the *Karpathy* splits used in the community (Karpathy & Li, 2015b). From the training data, we build a set of 8 models using 'resnet18' and 'resnet50' architectures (He et al., 2016). To ensure some diversity, we start from pretrained models from either ImageNet (Deng et al., 2009) or Places365 (Zhou et al., 2017). Each model has its last fully-connected ('fc') linear layer replaced by a linear layer allowing for 80 output categories. All these pretrained models are fine-tuned with some variations: The 'fc' layer is trained for all models, some also fine-tune the rest of the model, while some fine-tune only the 'layer4' of the ResNet architecture. These variations are summarized in Table 5. Training of the 'fc' layer uses a $10^{-3}$ learning rate, while all fine-tunings use $10^{-6}$ learning rate. All multi-label trainings use ADAM (Kingma & Ba, 2015) with ($\beta_1 = 0.9, \beta_2 = 0.999$) for learning rate management and are stopped at 40 epochs. Only the center crop of $224*224$ of an input image is used once its largest dimension is resized to 256.

| Architecture | Pretraining | Training | | |
| --- | --- | --- | --- | --- |
| | | fc only | fc + fine-tuning | fc + layer4 fine-tuning |
| resnet18 | ImageNet | r18.img.fc | r18.img.fc+ft | - |
| | Places365 | r18.plc.fc | r18.plc.fc+ft | - |
| resnet50 | ImageNet | r50.img.fc | - | - |
| | Places365 | r50.plc.fc | r50.plc.fc+ft | r50.plc.fc+ft4 |

Table 5: Description of our 8 models built on MS-COCO

**Evaluation Metric.** We use the mean Average Precision (mAP) which gives the area under the curve of $P = f(R)$ for precision $P$ and recall $R$, averaged over each class. mAP performs a sweep of the threshold used for detecting a positive class and captures a broad view of a multi-label predictor performance. Performances for our 8 models are reported in Table 6. Precision, Recall and F1 for micro/macro are given in Table 10. Our individual models have reasonable performances overall.

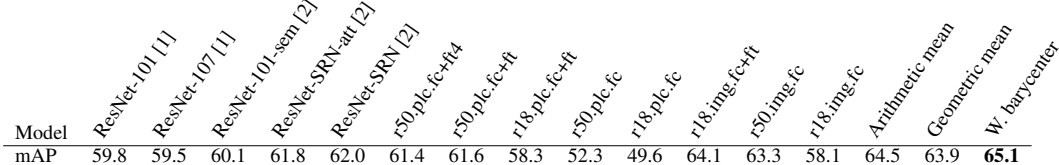

| Model | ResNet-101 [1] | ResNet-107 [1] | ResNet-101-sem [2] | ResNet-SRN-att [2] | ResNet-SRN [2] | r50.plc.fc+ft4 | r50.plc.fc+ft | r18.plc.fc+ft | r50.plc.fc | r18.plc.fc | r18.img.fc+ft | r50.img.fc | r18.img.fc | Arithmetic mean | Geometric mean | W. barycenter |
| --- | --- | --- | --- | --- | --- | --- | --- | --- | --- | --- | --- | --- | --- | --- | --- | --- |
| mAP | 59.8 | 59.5 | 60.1 | 61.8 | 62.0 | 61.4 | 61.6 | 58.3 | 52.3 | 49.6 | 64.1 | 63.3 | 58.1 | 64.5 | 63.9 | **65.1** |

Table 6: Multi-label models performances compared to published results on MS-COCO test set. W. barycenter outperforms arithmetic & geometric means. [1] He et al. (2015) [2] Zhu et al. (2017)

Arithmetic and geometric means offer direct mAP improvements over our 8 individual models. For unbalanced W. barycenter, the transport of probability mass is completely defined by its matrix $K = K_\ell$ in Algorithm 2. We investigated multiple $K$ matrix candidates by defining $K(i, j)$ as (i) the pairwise GloVe distance between categories, (ii) pairwise visual word2vec embeddings distance, (iii) pairwise co-occurence counts from training data. In our experience, it is challenging to find a generic $K$ that works well overall. Indeed, W. barycenter will move mass exactly as directed by $K$. A generic $K$ from prior knowledge may assign mass to a category that may not be present in some images at test time, and get harshly penalized by our metrics. A successful approach is to build a diagonal $K$ for each test sample based on the top-N scoring categories from each model and assign the average of model posteriors scores $K(i, i) = \frac{1}{M} \sum_m p_m(i|x)$ for image $x$ and category $i$. If a category is not top scoring, a low $K(i, i) = \zeta$ value is assigned to it, diminishing its contribution. It gives W. barycenter the ability to suppress categories not deemed likely to be present, and reinforce the contributions of categories likely to be. This simple diagonal $K$ gives our best results when using the top-2 scoring categories per model (the median number of active class in our training data is about 2) and outperforms arithmetic and geometric means as seen in Table 6. In all our experiments, W. barycenters parameters $\{\varepsilon, \lambda\}$ in Algorithm 2 and $\zeta$ defined above were tuned on validation set (5K). We report results on MS-COCO test set ($\approx$35K). In this task of improving our 8 models, W. barycenter offers a solid alternative to commonly used arithmetic and geometric means. Appendix B.2 shows that non-uniform weighting further improves W. ensembling performance.

## 6.3 IMAGE CAPTIONING

In this task the objective is to find a semantic consensus by ensembling 5 image captioner models. The base model is an LSTM-based architecture augmented with the attention mechanism over the image. In this evaluation we selected captioners trained with cross entropy objective as well as GAN-trained models (Dognin et al., 2018). The training was done on COCO dataset (Lin et al., 2014) using data splits from (Karpathy & Li, 2015a): training set of 113k images with 5 captions each, 5k validation set, and 5k test set. The size of the vocabulary size is 10096 after pruning words with counts less than 5. The matrix $K_\ell = K$ in Algorithm 1 was constructed using word similarities, defined based on (i) GloVe word embeddings, so that $K = \exp(-C/\varepsilon)$, where cost matrix $C$ is constructed based on euclidean distance between normalized embedding vectors; and (ii) synonym relationships, where we created $K$ based on the word synonyms graph and user votes from Power Thesaurus. The model prediction $\mu_\ell$, for $\ell = 1, \ldots, 5$ was selected as the softmax output of the captioner's LSTM at the current time step, and each model's input was weighted

equally: $\lambda_\ell = 1/m$. Once the barycenter $p$ was computed, the result was fed into a beam search (beam size $B = 5$), whose output, in turn, was then given to the captioner's LSTM and the process continued until a stop symbol (EOS) was generated. In order to exploit the controllable entropy of W. barycenter via the entropic regualrization parameter $\varepsilon$, we also decode using randomized Beam search of (Shao et al., 2017), where instead of maintaining the top k values, we sample $D$ candidates in each beam. The smoothness of the barycenter in semantic clusters and its controllable entropy promotes diversity in the resulting captions. We baseline the W. barycenter ensembling with arithmetic and geometric means.

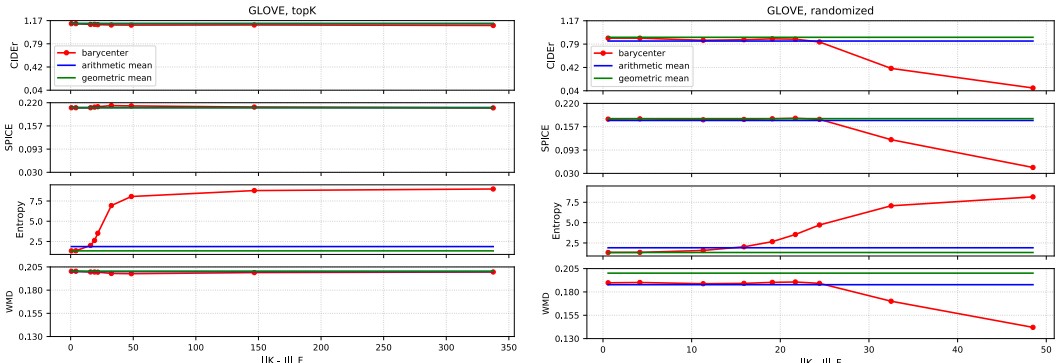

Figure 1: Comparison of the ensembling methods on COCO validation set using GloVe-based similarity matrix $K$ for 2 versions of beam search: topK (left panel) and randomized (right panel). The x-axis shows $\|K - I\|_F$, which corresponds to a different regularization parameter $\varepsilon$ (varied form 1 to 50). We can see that for topK beam search (left panel) the further $K$ is from the identity matrix, the larger the similarity neighborhood of each word, the more diverse are the generated captions (the barycenter has higher entropy), while still remaining semantically close to the ground truth. On the other hand, for randomized beam search (right panel), it is important to maintain a smaller similarity neighborhood, so that the generated sentences are not too different from the referenced ground truth.

**Controllable entropy and diversity.** Figures 1 and 2 show the comparison of the ensembling methods on the validation set using topK and randomized beam search. The x-axis shows $\|K - I\|_F$, which corresponds to a different regularization $\varepsilon$ (varied form 1 to 50). We report two n-gram based metrics: CIDEr and SPICE scores, as well as the WMD (Word Mover Distance) similarity (Kusner et al., 2015), which computes the earth mover distance (the Wasserstein distance) between the generated and the ground truth captions using the GloVe word embedding vectors.

In topK beam search, as $\varepsilon$ increases, causing the entropy to go up, the exact n-grams matching metrics, i.e., CIDEr and SPICE, deteriorate while WMD remains stable. This indicates that while the barycenter-based generated sentences do not match exactly the ground truth, they still remain semantically close to it (by paraphrasing), as indicated by the stability of WMD similarity. The results of the GloVe-based barycenter on the test split of COCO dataset are shown in Table 7. In randomized beam search, the increase in entropy of the barycenter leads to a similar effect of paraphrasing but this works only up to a smaller value of $\varepsilon$, beyond which we observe a significant deterioration of the results. At that point all the words become neighbors and result in a very diffused barycenter, close to a uniform distribution. This diffusion effect is smaller for the synonyms-based $K$ since there are only a certain number of synonyms for each word, thus the maximum neighborhood is limited.

|  | CIDER | SPICE | Entropy | WMD |
|---|---|---|---|---|
| Barycenter $\|K - I\|_F = 48.5$ | 1.091 | 0.209 | 6.94 | 0.198 |
| Geometric | 1.119 | 0.203 | 1.33 | 0.200 |
| Arithmetic | 1.117 | 0.205 | 1.87 | 0.200 |

Table 7: Performance of GloVe-based W. barycenter on COCO test split using topK beam search versus Geometric and Arithmetic ensembling. While the generated sentences based on W. barycenter do not match exactly the ground truth (lower CIDEr), they remain semantically close to it, while being more diverse (e.g., paraphrased) as indicated by the higher entropy and stable WMD.

**Robustness of W. Barycenter to Semantic Perturbations.** Finally, the right panel of Figure 2, shows the robustness of the W. barycenter to random shuffling of the $\mu_\ell$ values, within semantically coherent clusters. Note that the size of those clusters increases as $K$ moves away from identity. The results show that barycenter is able to recover from those perturbations, employing the side information from $K$, while both the arithmetic and geometric means (devoid of such information) are confused by this shuffling, displaying a significant drop in the evaluation metrics.

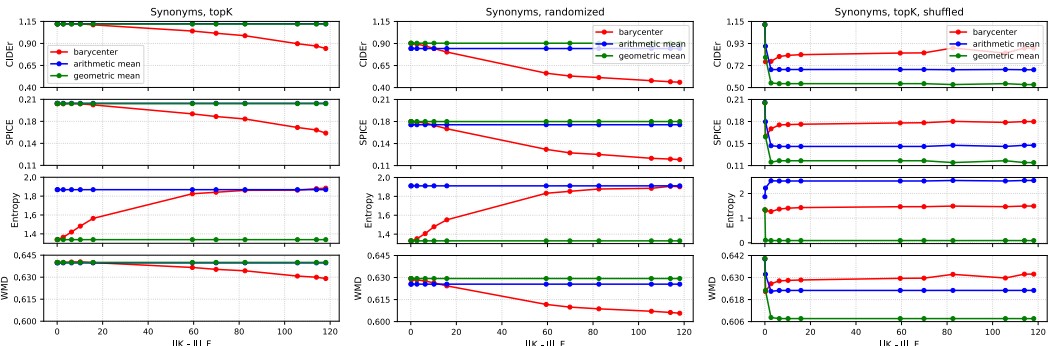

Figure 2: Left and Center Panels: Comparison of the ensembling methods on COCO validation set using synonyms-based similarity matrix with topK and randomized beam search. Right Panel: Comparison of ensembling methods when the predictions of the input models are shuffled according to the neighborhood structure defined by $K$. It can be seen that the W. Barycenter ensembling is able to recover from the word shuffling and produce better captions then the simple averaging methods, which are not able to exploit the provided side information.

**Human Evaluation.** We performed human evaluation on Amazon MTurk on a challenging set of images out of context of MS-COCO (Dognin et al., 2018). We compared three ensembling techniques: arithmetic, geometric and W. barycenter. For W. barycenter we used the similarity matrix $K$ defined by visual word2vec (Kottur et al., 2016). For the three models we use randomized beam search. We asked MTurkers to give a score for each caption on a scale 1-5 and choose the best captions based on correctness and detailedness. Captions examples are given in Fig. 6 (Appendix). Fig. 3 shows that W. barycenter has an advantage over the basic competing ensembling techniques.

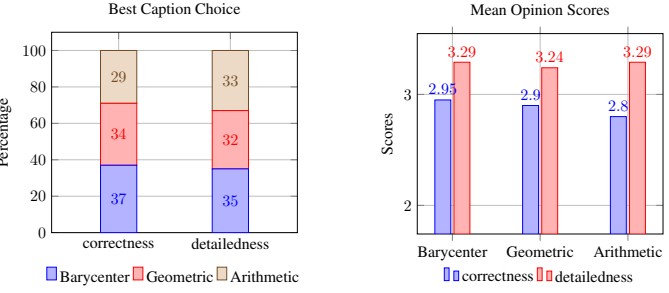

Figure 3: Human evaluation results. Left: Percentage of human picking best captions in terms of correctness and detail. Right: Mean Opinion Score on a scale 1 to 5.

# 7 CONCLUSION

We showed in this paper that W. barycenters are effective in model ensembling in machine learning. In the unbalanced case we showed their effectiveness in attribute based classification, as well as in improving the accuracy of multi-label classification. In the balanced case, we showed that they promote diversity and improve natural language generation by incorporating the knowledge of synonyms or word embeddings.

ACKNOWLEDGMENTS

Authors would like to thank Tommi Jaakkola, David Alvarez-Melis and Rogerio Feris for fruitful discussions. Authors thank also Satwik Kottur for sharing his visual word embeddings.

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

## A   IMAGE CAPTIONING

In this Section we provide additional results for evaluating the W. barycenter on image captioning task. Figure 4 (which corresponds to Table 2 of the main paper), visualizes the word distribution of the considered ensembling methods for the input of 4 models. It is clear that with a proper choice of similarity matrix $K$, W. barycenter can create diverse, high entropy outputs.

| Rank | 48.5 ($\varepsilon = 10$) | | 39.2 ($\varepsilon = 9$) | | 27.8 ($\varepsilon = 7$) | | 21.7 ($\varepsilon = 5$) | | 15.9 ($\varepsilon = 3$) | | 4.2 ($\varepsilon = 1$) | |
|---|---|---|---|---|---|---|---|---|---|---|---|---|
| 0 | car | 08.09 | car | 13.63 | car | 30.39 | car | 40.43 | car | 41.87 | car | 41.94 |
| 1 | cars | 00.99 | cars | 01.36 | cars | 01.85 | truck | 02.30 | truck | 02.22 | truck | 02.23 |
| 2 | vehicle | 00.63 | truck | 00.88 | truck | 01.81 | black | 01.60 | black | 01.67 | black | 01.67 |
| 3 | truck | 00.60 | vehicle | 00.86 | vehicle | 01.30 | cars | 01.52 | train | 01.51 | train | 01.51 |
| 4 | van | 00.40 | van | 00.59 | black | 01.12 | train | 01.46 | fashion | 01.44 | fashion | 01.49 |
| 5 | automob | 00.38 | automob | 00.45 | van | 00.94 | vehicle | 01.31 | bus | 01.30 | bus | 01.30 |
| 6 | black | 00.26 | black | 00.44 | train | 00.88 | bus | 01.20 | vehicle | 01.14 | vehicle | 01.14 |
| 7 | bus | 00.26 | bus | 00.39 | bus | 00.83 | van | 01.01 | photo | 01.01 | photo | 01.01 |
| 8 | parking | 00.24 | parking | 00.37 | parking | 00.79 | parking | 00.98 | van | 01.01 | van | 01.01 |
| 9 | vehicle | 00.23 | train | 00.32 | photo | 00.65 | photo | 00.96 | red | 01.00 | red | 01.01 |
| 10 | passeng | 00.21 | passeng | 00.32 | passeng | 00.59 | red | 00.90 | cars | 00.95 | parking | 00.94 |
| 11 | train | 00.20 | vehicle | 00.27 | red | 00.53 | fashion | 00.85 | parking | 00.94 | buildin | 00.88 |
| 12 | auto | 00.19 | photo | 00.27 | white | 00.49 | white | 00.75 | buildin | 00.88 | cars | 00.81 |
| 13 | driving | 00.19 | red | 00.21 | automob | 00.44 | buildin | 00.72 | passeng | 00.71 | passeng | 00.71 |
| 14 | photo | 00.17 | white | 00.21 | model | 00.35 | passeng | 00.70 | white | 00.70 | white | 00.67 |
| 15 | suv | 00.16 | auto | 00.21 | buildin | 00.35 | model | 00.54 | model | 00.59 | model | 00.60 |
| 16 | red | 00.14 | suv | 00.21 | silver | 00.32 | silver | 00.45 | picture | 00.48 | picture | 00.49 |
| 17 | white | 00.14 | driving | 00.21 | pickup | 00.30 | city | 00.44 | silver | 00.47 | silver | 00.47 |
| 18 | taxi | 00.13 | model | 00.17 | vehicle | 00.29 | picture | 00.38 | style | 00.43 | style | 00.43 |
| 19 | pickup | 00.11 | pickup | 00.17 | suv | 00.29 | style | 00.37 | city | 00.38 | city | 00.38 |

Table 8: Sample output (top 20 words) of barycenter for different similarity matrices $K$ based on GloVe (columns titles denote the distance of $K$ from identity $\|K - I\|_F$ and corresponding $\epsilon$.). Each column shows a word and its corresponding probability over the vocabulary. Note that the last column coincides with the output from geometric mean.

Table 8 shows the effect of entropic regularization $\varepsilon$ on the resulting distribution of the words of W. barycenter using GloVe embedding matrix. As $K$ moves closer to the identity matrix, the entropy of barycenter decreases, leading to outputs that are close/identical to the geometric mean. On the other hand, with a large entropic regularization, matrix $K$ moves away from identity, becoming an uninformative matrix of all 1's. This eventually leads to a uniform distribution which spreads the probability mass equally across all the words. This can be also visualized with a histogram in Figure 5, where the histograms on the bottom represent distributions that are close to uniform, which can be considered as failure cases of W. barycenter, since the image captioner in this case can only generate meaningless, gibberish captions.

In Table 9 we show a mapping from a few top words in the barycenter output (for similarity matrix $K$ based on synonyms) to the input models. In other words, each column defines the words in the input models which have the greatest influence on each of top 3 words in the barycenter output.

In Figure 6 we present a few captioning examples showing qualitative difference between the considered ensembling techniques.

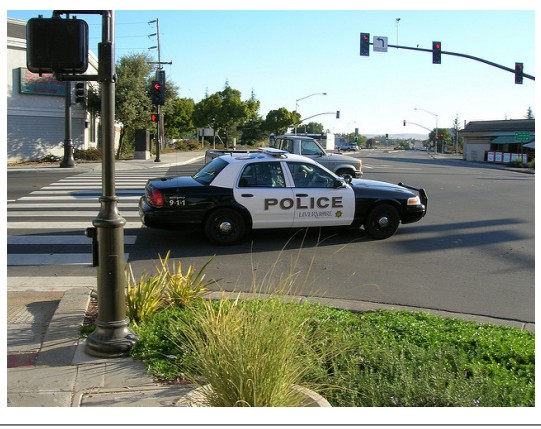

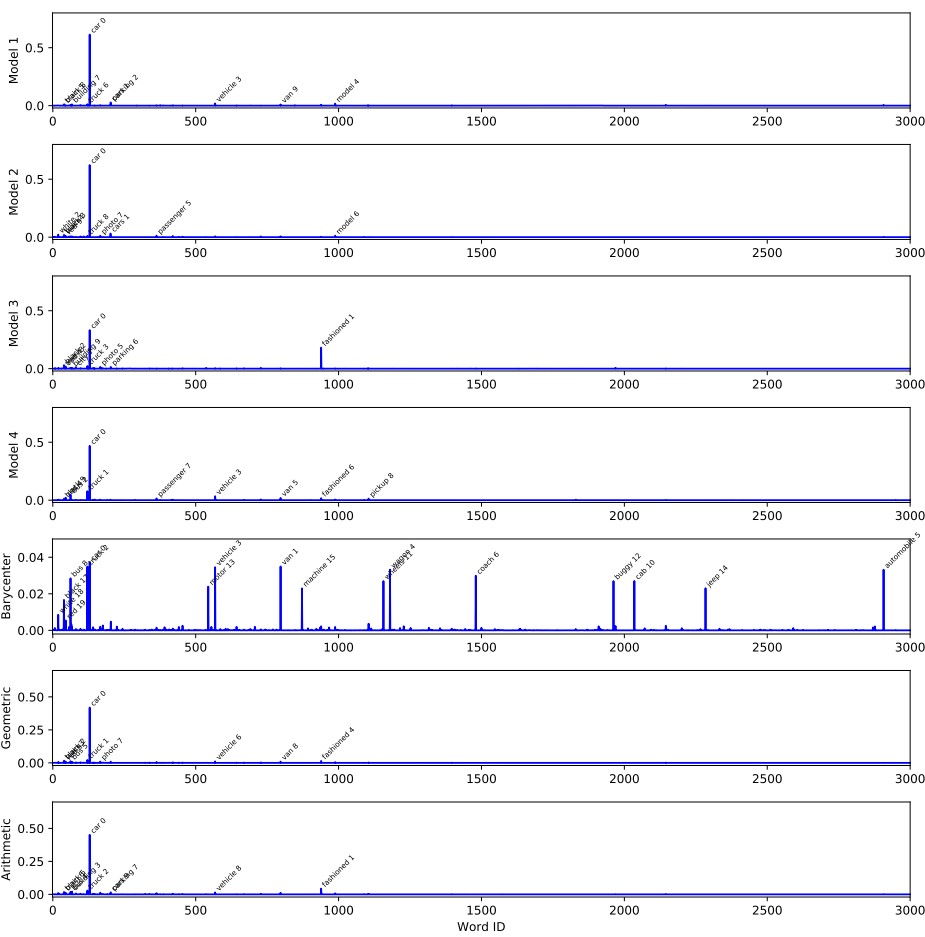

Figure 4: Visualization of the word distributions of W. barycenter, arithmetic and geometric means based on four captioning models, whose input image is shown on top (one of the ground-truth human-annotated captions for this image reads: *A police car next to a pickup truck at an intersection*). The captioner generates a sentence as a sequence of words, where at each step the output is a distribution over the whole vocabulary. The top four histograms show a distribution over the vocabulary from each of the model at time $t = 3$ during the sentence generation process. The bottom three histograms show the resulting distribution over the vocabulary for the ensembles based on W. Barycenter, arithmetic and geometric means. It can be seen that the W. Barycenter produces high entropy distribution, spreading the probability mass over the synonyms of the word "car" (which is the top word in all the four models), based on the synonyms similarity matrix $K$.

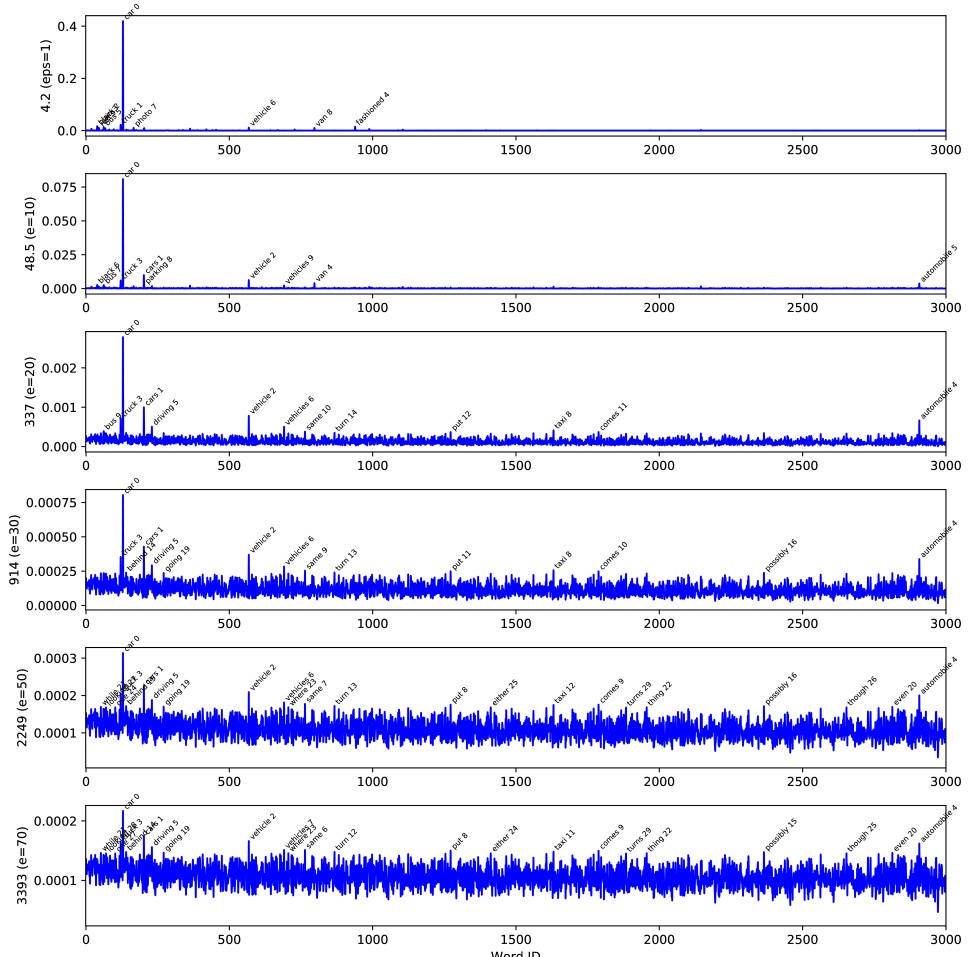

Figure 5: Visualization of the word distributions of W. barycenter for different similarity matrices $K$ based on GloVe (rows denote the distance of $K$ from identity $\|K - I\|_F$ and corresponding $\epsilon$). Large entropic regularization generates $K$ close to uninformative matrices of all 1's. This eventually leads to a barycenter which is close to a uniform distribution spreading the probability mass almost equally across all the words.

| Word | Model 1 | | Model 2 | | Model 3 | | Model 4 | |
|---|---|---|---|---|---|---|---|---|
| | car | 90.00 | car | 95.28 | car | 84.96 | car | 53.93 |
| | vehicle | 5.16 | vehicle | 1.33 | truck | 9.08 | truck | 20.67 |
| car | van | 1.62 | bus | 1.26 | bus | 2.74 | vehicle | 10.75 |
| | automobile | 0.92 | van | 0.93 | vehicle | 1.41 | bus | 10.01 |
| | bus | 0.85 | truck | 0.75 | van | 0.88 | van | 3.86 |
| | car | 97.89 | car | 99.46 | car | 97.60 | car | 97.46 |
| | automobile | 1.30 | automobile | 0.38 | motorcycle | 1.72 | motorcycle | 1.28 |
| jeep | jeep | 0.51 | jeep | 0.08 | jeep | 0.28 | jeep | 0.64 |
| | motorcycle | 0.27 | motorcycle | 0.07 | cab | 0.23 | cab | 0.46 |
| | limousine | 0.02 | cab | 0 | automobile | 0.16 | automobile | 0.16 |
| | silver | 53.11 | white | 95.61 | white | 88.27 | white | 82.68 |
| | white | 46.49 | silver | 4.37 | snow | 6.63 | silver | 17.18 |
| white | snowy | 0.30 | snowy | 0.02 | silver | 4.66 | snowy | 0.12 |
| | pale | 0.06 | pale | 0 | pale | 0.24 | pale | 0.01 |
| | blank | 0.04 | blank | 0 | blank | 0.2 | ivory | 0.01 |

Table 9: Mapping from a few top words in the barycenter output (for similarity matrix $K$ based on synonyms) to the input models. For each word in the left columns, the remaining columns show the contributing words and the percent of contribution.

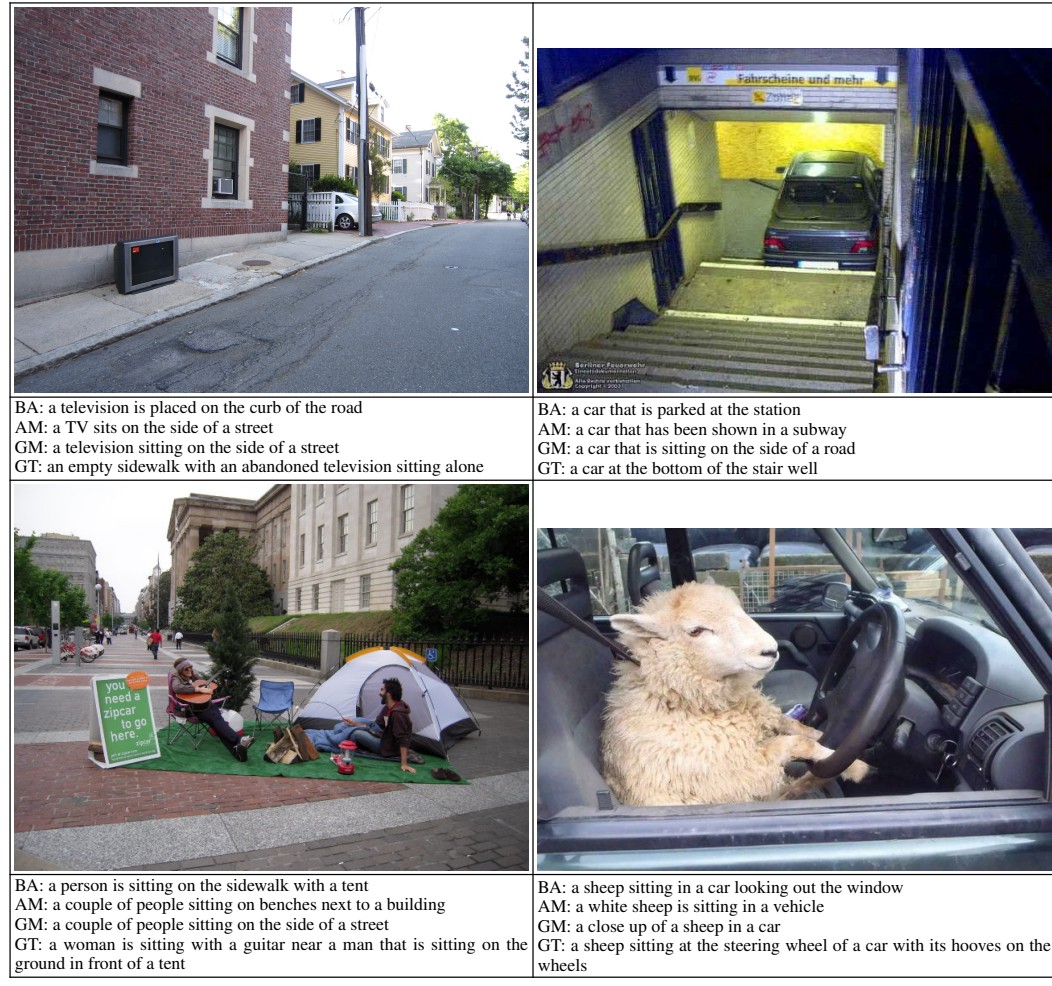

BA: a television is placed on the curb of the road
AM: a TV sits on the side of a street
GM: a television sitting on the side of a street
GT: an empty sidewalk with an abandoned television sitting alone

BA: a car that is parked at the station
AM: a car that has been shown in a subway
GM: a car that is sitting on the side of a road
GT: a car at the bottom of the stair well

BA: a person is sitting on the sidewalk with a tent
AM: a couple of people sitting on benches next to a building
GM: a couple of people sitting on the side of a street
GT: a woman is sitting with a guitar near a man that is sitting on the ground in front of a tent

BA: a sheep sitting in a car looking out the window
AM: a white sheep is sitting in a vehicle
GM: a close up of a sheep in a car
GT: a sheep sitting at the steering wheel of a car with its hooves on the wheels

Figure 6: Examples of captions for several images. BA: Wasserstein Barycenter, AM: Arithmetic mean, GM: Geometric mean, GT: Ground truth.

## B    MULTI-LABEL PREDICTION

### B.1    METRICS USED AND SINGLE MODELS PREDICTIONS

We evaluate our models using micro and macro versions of precision, recall, and F1-measure as covered in multi-label prediction metrics study from (Wu & Zhou, 2016). For these measures, a threshold of 0.5 is commonly used to predict a label as positive in the community's published results. Macro precision is an average of per-class precisions while micro precision is computed by computing the ratio of all true positives across all image samples over the number of all positive classes in a dataset. Therefore a macro (or *per-class*) precision 'P-C' is defined as $\frac{1}{C} \sum_i P_i$ while a micro (or *overall* precision) 'P-O' is defined as $\frac{\sum_i TP_i}{\sum_i TP_i + FP_i}$ where $TP_i$ and $FP_i$ are true and false positives respectively. Per-class and overall versions for R and F1 are defined similarly. We also employ mean Average Precision (mAP) which gives the area under the curve of $P = f(R)$ averaged over each class. Unlike P,R and F1, mAP inherently performs a sweep of the threshold used for detecting a positive class and captures a broader view of a multi-label predictor's performance. Performances for our 8 models and previously published results are reported in Table 10 and in Table 6 in the paper. Our models have reasonable performances overall.

| Model | mAP | F1-C | P-C | R-C | F1-O | P-O | R-O |
|---|---|---|---|---|---|---|---|
| ResNet-101 [1] | 59.8 | 55.7 | 65.8 | 51.9 | 72.5 | 75.9 | 69.5 |
| ResNet-107 [2] | 59.5 | 55.6 | 65.4 | 52.2 | 72.6 | 75.5 | 70.0 |
| ResNet-101-sem [2] | 60.1 | 54.9 | 69.3 | 48.6 | 72.6 | 76.9 | 68.8 |
| ResNet-SRN-att [2] | 61.8 | 56.9 | 67.5 | 52.5 | 73.2 | 76.5 | 70.1 |
| ResNet-SRN [2] | 62.0 | 58.5 | 65.2 | 55.8 | 73.4 | 75.5 | 71.5 |
| r50.plc.fc+ft4 | 61.4 | 54.6 | 74.9 | 43.0 | 63.4 | 81.3 | 51.9 |
| r50.plc.fc+ft | 61.6 | 55.5 | 74.2 | 44.4 | 64.1 | 80.7 | 53.2 |
| r18.plc.fc+ft | 58.3 | 51.0 | 71.9 | 39.5 | 61.2 | 80.0 | 49.6 |
| r50.plc.fc | 52.3 | 43.8 | 71.3 | 31.6 | 55.6 | 79.8 | 42.7 |
| r18.plc.fc | 49.6 | 40.7 | 68.9 | 28.9 | 54.1 | 78.8 | 41.2 |
| r18.img.fc+ft | 64.1 | 57.9 | 76.7 | 46.4 | 64.3 | 80.9 | 53.4 |
| r50.img.fc | 63.3 | 55.9 | 78.7 | 43.4 | 62.2 | 83.1 | 49.7 |
| r18.img.fc | 58.1 | 50.7 | 75.7 | 38.1 | 58.4 | 80.5 | 45.8 |

Table 10: Our multi-label models performances compared to published results on MS-COCO test set. Arithmetic, geometric means and W. barycenter performances are reported as well. [1] (He et al., 2015) [2] (Zhu et al., 2017)

## B.2 WEIGHTED MULTI-LABEL PREDICTION ENSEMBLING

Ensembling results given in Tab. 6 are using uniformly weighted models, i.e. $\lambda_\ell = \frac{1}{m}$ where $m$ is the number of models. However, in practice, arithmetic and geometric mean ensembling usually use weighted ensembles of models The weights are then optimized and established on a small validation set before being used for ensembling on a test set. A well-known embodiment of this type of approach is Adaboost (Freund & Schapire, 1999) where weights are dynamically defined at each pass of training wrt to the accuracy of base models.

Here, we follow a much simpler but similar approach by defining the performance of each model $\ell$ as the mean average precision ($\text{mAP}_\ell$) on the validation set. $\text{mAP}_\ell$ is used to define $\lambda_\ell$ such that $\lambda_\ell = \frac{\text{mAP}_\ell}{\sum_\ell \text{mAP}_\ell}$. $\lambda_\ell$ are then applied to the models' scores for arithmetic, geometric mean and W.Barycenter ensemblings. Tab. 11 reports mAP for each ensembling technique over the MS-COCO test set (35150 images). Note that the $\lambda_\ell$ weights definition is based on the final metric evaluation, mAP in this case. For other tasks such as classification, accuracy or any other valid metric can be employed to compute the $\lambda_\ell$ weights. It must be noted that the weights are computed with respect to the ultimate performance metric at hand. Tab. 11 reveals clearly that such approach of weighting models by their performance benefits arithmetic and W.Barycenter ensembling for this task. Both methods leverage the confidence of the underlying models and the mAP weighting of models will reinforce the contributions of better performing models. Geometric means ensembling is not significantly impacted by non-uniform $\lambda_\ell$ since it is mostly relying on consensus of the models, not their confidence. We conclude that weighting indeed helps performance and keeps a performance advantage for W. Barycenter over the alternatives arithmetic and geometric means.

| Ensembling | $\lambda_\ell = \frac{1}{m}$ | $\lambda_\ell = \frac{\text{mAP}_\ell}{\sum_\ell \text{mAP}_\ell}$ |
|---|---|---|
| Arithmetic mean | 64.5 | 64.8 |
| Geometric mean | 63.9 | 63.9 |
| W.Barycenter | **65.1** | **65.2** |

Table 11: multi-label models ensembling mAP on MS-COCO test set (35150 images). Performance-based weighting helps both arithmetic and W.Barycenter ensembling, the latter retaining its performance vantage.

## C    OTHER MACHINE LEARNING TASKS THAT CAN BE ADRESSED IN THE WASSERSTEIN BARYCENTER ENSEMBLING FRAMEWORK

Table 12 summarizes Machine learning tasks that could benefit from W. Ensembling, especially in the case when source and target domains are distinct.

| | Source Domains (models) | Target Domain (Barycenter) | Kernel $K$ (OT cost matrix) | Arithmetic Geometric apply |
|---|---|---|---|---|
| Multi-class Learning (Balanced OT) | $\mu_\ell \in \Delta_N, \ell = 1 \ldots m$ Histograms of size $N$ | $p \in \Delta_N$ Histogram of size N | $K_{ij} = e^{-\|x_i - x_j\|^2}$ $x_i$ word embedding of category $i$ (See Section 6.3 for e.g GloVe or Visual w2v) | ✓ |
| Multi-label Learning (Unbalanced OT) | $\mu_\ell \in [0,1]^N, \ell = 1 \ldots m$ Soft multi-labels | $p \in [0,1]^N$ Soft multi-label | $K_{ij}$ adjacency weight in a knowledge graph $K_{ij}$= co-occurrence of item $i$ and $j$ (See Section 6.2 for other choices) | ✓ |
| Cost-sensitive Classification (Balanced OT) | $\mu_\ell \in \Delta_N, \ell = 1 \ldots m$ Histograms of size $N$ | $p \in \Delta_N$ Histogram of size $N$ | $K_{ij}$ User ratings of similarities user-defined costs for confusion of $i$ and $j$ e.g: binary matrix for synonyms (Section 6.3) | ✓ |
| Attribute to Categories Zero (Few) shot Learning (Unbalanced OT) | $\mu_\ell \in [0,1]^N, \ell = 1 \ldots m$ Soft multi-labels: $N$ attributes | $p \in \Delta_M$ Histogram of size $M$ $M$ categories | $K_{ij}$ presence or absence of attribute $i$ in class $j$ (See Section 6.1) | ✗ |
| Vocabulary Expansion (Balanced OT) | $\mu_\ell \in \Delta_N, \ell = 1 \ldots m$ Histograms on a vocabulary of size $N$ | $p \in \Delta_M, M > N$ Histogram on a larger vocabulary | $K_{ij} = e^{-\frac{\|x_i - x_j\|^2}{\varepsilon}}$, $x_i$ word embeddings | ✗ |
| Multi-lingual fusion and Translation (Balanced OT) | $\mu_\ell \in \Delta_{N_\ell}, \ell = 1 \ldots m$ Histograms on $m$ source languages of vocabulary size $N_\ell$ each | $p \in \Delta_M$ Histogram on a target language with vocabulary size $M$ | $K_{\ell,ij} = e^{-\frac{\|x_i^\ell - y_j\|^2}{\varepsilon}}$, $x_i^\ell, y_j$ multi-lingual word embeddings: $x_i^\ell$ word embeddings of source language $\ell$ $y_j$ word embeddings of target language | ✗ |

Table 12: Machine learning tasks where W. Barycenter ensembling can be applied: We emphasize that W. Barycenter has the advantage over alternatives such as arithmetic or geometric means in that it is able to ensemble models whose histograms are defined on different domains. Moreover, the target domain, where the barycenter is defined, can be different from the source domains. This is encountered, for instance, in the attribute-based classification, where models are defined on attributes (multi-labels) and the ensemble is defined on the set of categories. We give here additional tasks that can benefit from this flexibility of W. Barycenters.

## D    CONVERGENCE OF SCALING ALGORITHMS FOR ENTROPIC REG. W. BARYCENTERS

For W. Barycenters (Chizat et al., 2018) proves (in Theorem 4.1 of this paper ) that the fixed point algorithm Given in Algorithms 1 and 2 converges under mild conditions on the matrix $K$: ($K$ takes positive values), nevertheless this does not characterize the convergence as $\varepsilon$ goes to zero.

### D.1    BALANCED W. BARYCENTERS

$\Gamma$ convergence of the regularized OT problem to the original OT problem when the regularization parameter vanishes was studied in (Carlier et al., 2017), but this work does provide an algorithm that guarantees such a convergence.

For balanced W. barycenters we show in the following Lemma that the fixed point of algorithm 1 converges to the geometric mean when $K_\ell = I$ for all $\ell = 1 \ldots m$. Assuming $C_{ij}$ is bounded, it is easy to see that $K$ converges to identity as $\varepsilon$ approaches 0. Hence the fixed point of Algorithm 1 does not recover the original unregularized W. Barycenter as $\varepsilon \to 0$.

**Lemma 1.** *For $K_\ell = I, \ell = 1 \ldots m$, the fixed point of Algorithm 1 is the geometric mean $\bar\mu_g = \Pi_{\ell=1}^m (\mu_\ell)^{\lambda_\ell}$.*

*Proof.* Let us show that at any iteration we have the following recurrence:

$$u_\ell^t = \frac{\mu_\ell^{t+1}}{(\bar\mu_g)^t}, p^t = \bar\mu_g, v_\ell^t = \left(\frac{\bar\mu_g}{\mu_\ell}\right)^{t+1},$$

all operations are element-wise.

At iteration 0, the result holds since we have :

$$u_\ell^0 = \mu_\ell, p^0 = \Pi_{\ell=1}^m (\mu_\ell)^{\lambda_\ell}, v_\ell^0 = \frac{\Pi_{\ell=1}^m (\mu_\ell)^{\lambda_\ell}}{\mu_\ell}$$

Assume the result holds at time $t$. Let us prove it for $t+1$, following the updates of Algorithm 1 :

$$u_\ell^{t+1} = \frac{\mu_\ell}{v_\ell^t} = \frac{\mu_\ell}{\left(\frac{\bar{\mu}_g}{\mu_\ell}\right)^{t+1}} = \frac{(\mu_\ell)^{t+2}}{(\bar{\mu}_g)^{t+1}}$$

$$
\begin{aligned}
p^{t+1} &= \Pi_{\ell=1}^m \left(u_\ell^{t+1}\right)^{\lambda_\ell} \\
&= \Pi_{\ell=1}^m \frac{(\mu_\ell)^{(t+2)\lambda_\ell}}{(\bar{\mu}_g)^{(t+1)\lambda_\ell}} \\
&= \frac{\left(\Pi_{\ell=1}^m \mu_\ell^{\lambda_\ell}\right)^{t+2}}{(\bar{\mu}_g)^{(t+1)\sum_{\ell=1}^m \lambda_\ell}} \\
&= \frac{\bar{\mu}_g^{t+2}}{\bar{\mu}_g^{t+1}} \text{ since } \sum_{\ell=1}^m \lambda_\ell = 1 \\
&= \bar{\mu}_g.
\end{aligned}
$$

$$v_\ell^{t+1} = \frac{p}{u_\ell^{t+1}} = \frac{\bar{\mu}_g}{\frac{(\mu_\ell)^{t+2}}{(\bar{\mu}_g)^{t+1}}} = \left(\frac{\bar{\mu}_g}{\mu_\ell}\right)^{t+2}.$$

QED. □

For $\varepsilon > 0$, Feydy et al (Feydy et al., 2018) showed recently that the Sinkhorn divergence defines an interpolation between the *MMD* distance (Maximum mean discrepancy (Gretton et al., 2012)) and the Wasserstein Distance.

Hence for $\varepsilon > 0$ Algorithm 1 provides still an interesting solution that can be seen as an interpolation between the original (unregularized) Wasserstein Barycenter and the MMD Barycenter (Frechet barycenter for $d = $ MMD).

## D.2 Unbalanced W. Barycenter

Note that

$$\min_\gamma \frac{1}{\lambda} \langle C, \gamma \rangle + \widetilde{\mathrm{KL}}(\gamma 1_M, p) + \widetilde{\mathrm{KL}}(\gamma^\top 1_N, q) - \frac{\varepsilon}{\lambda} H(\gamma)$$

$$= \min_\gamma \widetilde{\mathrm{KL}}(\gamma 1_M, p) + \widetilde{\mathrm{KL}}(\gamma^\top 1_N, q) + \frac{1}{\lambda}\left(\widetilde{\mathrm{KL}}(\gamma, e^{-C/\varepsilon}) - \varepsilon \sum_{i,j} K_{ij}\right)$$

As $\lambda$ goes to infinity this unbalanced cost converges to the Hellinger distance:

$$\min_\gamma \widetilde{\mathrm{KL}}(\gamma 1_M, p) + \widetilde{\mathrm{KL}}(\gamma^\top 1_N, q) = \|\sqrt{p} - \sqrt{q}\|^2,$$

Hence using $d = \frac{1}{\lambda} W_{unb,\varepsilon}$ in Unbalanced Barycenters, we approach the Hellinger Frechet mean, as $\lambda \to \infty$.

## E  Time Complexity of Scaling Algorithms for W. Barycenter

### E.1  Computational Complexity and Improvements

The total time complexity of a straight-forward implementation of Algorithms 1 and 2 is $O(mN^2\text{Maxiter})$ where Maxiter is the number of iterations, $m$ the number of models and $N$ the number of bins.

We make the following theoretical and practical remarks on how to improve this computational complexity to reach an almost linear dependency on $N$ using low rank approximation of the kernel matrix $K$, and parallelization on $m$ machines:

1. *Dependency on **Maxiter**:* For the number of iterations we found that Maxiter $=5$ is enough for convergence, which makes most of the computational complexity dependent on $m$ and $N$.

2. *Dependency on **N** and low rank approximation :* The main computational complexity comes from the matrix vector multiply $K_\ell u_\ell$ that is of $O(N^2)$. Note that this complexity can be further reduced since the kernel matrix $K$ is often low rank. Therefore we can be written $K = \Phi\Phi^\top$ where $\Phi \in \mathbb{R}^{N \times k}$, where $k \ll N$, which allows to compute this product as follows $\Phi\Phi^\top u_\ell$ that has a lower complexity $O(Nk)$. $\Phi$ can be computed using Nystrom approximation or random Fourier features. Hence potentially on can get an algorithm with complexity $O(mNk)$, where $k$ has a logarithmic dependency on $N$. This was studied recently in (Altschuler et al., 2018).

3. *Dependency on **m** and parallelization:* Regarding the dependency on $m$, as noted in (Chizat et al., 2018; Benamou et al., 2015) the algorithm is fully parallelizable which would lead to a computational complexity of $O(Nk)$ by using simply $m$ machines .

4. *GPU and Batch version:* Practically, the algorithm implemented takes advantage of matrix vector products' speed on GPU. The algorithm can be further accelerated by computing Sinkhorn divergences in batches as pointed in (Feydy et al., 2018).

### E.2 TIME COMPLEXITY IN THE MULTI-LABEL WASSERSTEIN ENSEMBLING

We evaluated the time complexity of the GPU implementation of Wasserstein Barycenters in *pytorch* on our multi-label prediction experiments using MS-COCO test set (35150 samples). Note that we used a vanilla implementation of Algorithm 2, i.e without parallelization, batching, or low rank approximation. Results and comments for these wall clock timings can be found in Tab. 13. As it can be observed, we need to use Maxiter $= 5$ on a GPU-V100 to reach below 4ms/image for Wasserstein ensembling. This is not a major overhead and can be further improved as discussed previously by using parallelization, batching and low rank approximation. In Table 13, each timing was done over the whole test set; each timing repeated 5 times. We report means and standard deviations of total wall clock times for ensembling 8 models. Last column on the right is the average timing per image (in ms) for W.Barycenter. The number of W.Barycenter iterations (Maxiter) was varied from 1, 5 and 10 to show its impact. We report timing numbers over two GPU architectures, NVIDIA Tesla K80 and V100. W.Barycenters leverage the GPU while Arithmetic and Geometric means do not. Timings are of the computations of the means themselves, no data fetching or data preparation is included in these timings. As expected, the wall clock time cost for W.Barycenters is several order of magnitude higher than for Arithmetic and Geometric means. The difference of GPU does not impact the Arithmetic and Geometric means as they do not use it in our implementation. The Barycenter computation see a speed up from K80 to V100 as V100 is much better at reducing wall time for longer number of iterations.

| GPU Model | Maxiter Iterations | Arithmetic Mean test set (s) | | Geometric Mean test set (s) | | W.Barycenter test set (s) | | W.Barycenter per image (ms) |
|---|---|---|---|---|---|---|---|---|
| Tesla-K80 | 1 | 0.015 | $\pm$.001 | 0.298 | $\pm$.002 | 38.954 | $\pm$0.487 | 1.108 |
| | 5 | 0.015 | $\pm$.000 | 0.297 | $\pm$.001 | 152.987 | $\pm$1.725 | 4.352 |
| | 10 | 0.015 | $\pm$.000 | 0.296 | $\pm$.002 | 302.254 | $\pm$2.470 | 8.599 |
| Tesla-V100 | 1 | 0.018 | $\pm$.002 | 0.297 | $\pm$.003 | 36.742 | $\pm$0.843 | 1.045 |
| | 5 | 0.016 | $\pm$.002 | 0.289 | $\pm$.009 | 135.950 | $\pm$5.897 | 3.868 |
| | 10 | 0.014 | $\pm$.000 | 0.278 | $\pm$.001 | 249.775 | $\pm$3.119 | 7.106 |

Table 13: Timings (in s) of Wasserstein Barycenter computation compared to Arithmetic and Geometric mean computations for the MS-COCO test set (35150 samples).

## F    ADDITIONAL THEORECTICAL RESULTS AND PROOFS

**Proposition 2** (propreties of Geometric Mean). *The following properties hold for geometric mean:*

1. **Geometric mean is the Frechet mean of $\widetilde{KL}$.** *The geometric mean is the Frechet mean with respect to the extended $\widetilde{KL}$ divergence:*

$$\bar{\mu}_g = \Pi_{\ell=1}^{m}(\mu_\ell)^{\lambda_\ell} = \arg\min_{\rho} L(\rho) := \sum_{\ell=1}^{m} \lambda_\ell \widetilde{KL}(\rho, \mu_\ell)$$

2. **Correctness guarantee of Geometric mean in $\widetilde{KL}$.** *Let $\nu$ be an oracle the geometric mean satisfies:*

$$\widetilde{KL}(\nu, \bar{\mu}_g) \leq \sum_{\ell=1}^{m} \lambda_\ell \widetilde{KL}(\nu, \mu_\ell),$$

**Proposition 3** (properties of Arithmetic Mean). *The following properties hold for geometric mean:*

1. **Arithmetic mean is the Frechet mean of $\ell_2$.**

$$\bar{\mu}_a = \sum_{\ell=1}^{m} \lambda_\ell \mu_\ell = \arg\min_{\rho} L(\rho) := \sum_{\ell=1}^{m} \lambda_\ell \|\rho - \mu_\ell\|^2$$

2. **Correctness guarantee of Arithmetic mean in $\ell_2$.** *Let $\nu$ be an oracle the arithmetic mean satisfies:*

$$||\nu - \bar{\mu}_g||_2^2 \leq 2\sum_{\ell=1}^{m} \lambda_\ell \|\nu - \mu_\ell\|^2 .$$

3. **Entropy** : *Strong convexity of negative entropy with respect to $\ell_1$ of arithmetic mean:*

$$H(\sum_{\ell} \lambda_\ell \mu_\ell) \geq \sum_{\ell=1}^{m} \lambda_\ell H(\mu_\ell) + \frac{1}{2}\sum_{\ell \neq k} \lambda_\ell \lambda_k \|\mu_\ell - \mu_k\|_1^2 .$$

*Proof of Proposition 2.* 1)

$$\min_{\rho} L(\rho) := \sum_{\ell=1}^{m} \lambda_\ell \widetilde{KL}(\rho, \mu_\ell) = \sum_{i=1}^{N}\sum_{\ell=1}^{m} \lambda_\ell \left( \rho_i \log\left(\frac{\rho_i}{\mu_{\ell,i}}\right) - \rho_i + \mu_{\ell,i} \right)$$

First order optimality condition:

$$\frac{\partial L}{\partial \rho_i} = \sum_{\ell=1}^{m} \lambda_\ell \log\left(\frac{\rho_i}{\mu_{\ell,i}}\right) = 0,$$

This gives us the result:

$$\rho_i = \Pi_{\ell=1}^{m}(\mu_{\ell,i})^{\lambda_\ell}.$$

2)

$$
\begin{aligned}
\widetilde{\mathrm{KL}}(\nu, \bar{\mu}_g) &= \sum_i \left( \nu_i \log\left( \frac{\nu_i}{\Pi_{\ell=1}^m (\mu_{\ell,i})^{\lambda_\ell}} \right) - \nu_i + \Pi_{\ell=1}^m (\mu_{\ell,i})^{\lambda_\ell} \right) \\
&= \sum_i \left( \nu_i \log\left( \Pi_{\ell=1}^m (\frac{\nu_i}{\mu_{\ell,i}})^{\lambda_\ell} \right) - \nu_i + \Pi_{\ell=1}^m (\mu_{\ell,i})^{\lambda_\ell} \right) \text{ using } \sum_{\ell=1}^m \lambda_\ell = 1 \\
&= \sum_{\ell=1}^m \lambda_\ell \left( \sum_i \nu_i \log(\frac{\nu_i}{\mu_{\ell,i}}) - \nu_i \right) + \Pi_{\ell=1}^m (\mu_{\ell,i})^{\lambda_\ell} \\
&= \sum_{\ell=1}^m \lambda_\ell \left( \sum_i \nu_i \log(\frac{\nu_i}{\mu_{\ell,i}}) - \nu_i + \mu_{\ell,i} \right) + \sum_i \left( \Pi_{\ell=1}^m (\mu_{\ell,i})^{\lambda_\ell} - \sum_{\ell=1}^m \lambda_\ell \mu_{\ell,i} \right) \\
&= \sum_{\ell=1}^m \lambda_\ell \widetilde{\mathrm{KL}}(\nu, \mu_\ell) + \sum_i \left( \Pi_{\ell=1}^m (\mu_{\ell,i})^{\lambda_\ell} - \sum_{\ell=1}^m \lambda_\ell \mu_{\ell,i} \right) \\
&\leq \sum_{\ell=1}^m \lambda_\ell \widetilde{\mathrm{KL}}(\nu, \mu_\ell),
\end{aligned}
$$

The last inequality follows from the arithmetic-geometric mean inequality (Jensen inequality):

$$
\Pi_{\ell=1}^m (\mu_{\ell,i})^{\lambda_\ell} - \sum_{\ell=1}^m \lambda_\ell \mu_{\ell,i} \leq 0.
$$

3)

$$
\left| \widetilde{\mathrm{KL}}(\nu, \bar{\mu}_g) - \sum_{\ell=1}^m \lambda_\ell \widetilde{\mathrm{KL}}(\nu, \mu_\ell) \right| \leq \|\bar{\mu}_g - \bar{\mu}_a\|_1
$$

$$
\varepsilon_\ell - \|\bar{\mu}_g - \bar{\mu}_a\|_1 \leq \widetilde{\mathrm{KL}}(\nu, \bar{\mu}_g) \leq \varepsilon_u + \|\bar{\mu}_g - \bar{\mu}_a\|_1
$$

□

*Proof of Proposition 1.* 1) By the triangle inequality we have for all $\ell$:

$$
W_2(\bar{\mu}_w, \nu) \leq W_2(\bar{\mu}_w, \mu_\ell) + W_2(\mu_\ell, \nu)
$$

Raising to the power 2 we have:

$$
W_2^2(\bar{\mu}_w, \nu) \leq (W_2(\bar{\mu}_w, \mu_\ell) + W_2(\mu_\ell, \nu))^2 \leq 2(W_2^2(\bar{\mu}_w, \mu_\ell) + W_2^2(\mu_\ell, \nu)),
$$

where we used $(a+b)^2 \leq 2(a^2 + b^2)$. Summing over all $\ell$, and using $\sum_{\ell=1}^m \lambda_\ell = 1$ we have:

$$
\begin{aligned}
W_2^2(\bar{\mu}_w, \nu) &\leq 2 \left( \sum_{\ell=1}^m \lambda_\ell W_2^2(\bar{\mu}_w, \mu_\ell) + \sum_{\ell=1}^m \lambda_\ell W_2^2(\mu_\ell, \nu) \right) \\
&\leq 2 \left( \sum_{\ell=1}^m \lambda_\ell W_2^2(\nu, \mu_\ell) + \sum_{\ell=1}^m \lambda_\ell W_2^2(\mu_\ell, \nu) \right) \\
&\quad ( \text{By Barycenter definition } \sum_{\ell=1}^m \lambda_\ell W_2^2(\bar{\mu}_w, \mu_\ell) \leq \sum_{\ell=1}^m \lambda_\ell W_2^2(\nu, \mu_\ell).) \\
&= 4 \sum_{\ell=1}^m \lambda_\ell W_2^2(\nu, \mu_\ell) (\text{Using the symmetry of } W_2).
\end{aligned}
$$

2) For a fixed base distribution $\mu$ the functional $W_2^2(., \mu)$ is convex along generalized geodesics (Santambrogio, May 2015). Given two distributions $\nu_1, \nu_2$, and $T_1$ and $T_2$, such that $T_{1,\#}\mu = \nu_1$ and $T_{2,\#}\mu = \nu_2$, we have:

$$
W_2^2((tT_1 + (1-t)T_2)\mu, \mu) \leq t W_2^2(\nu_1, \mu) + (1-t) W_2^2(\nu_2, \mu).
$$

Fix $k \in [[1, m]]$. Note that there exists $T_\ell^k, \ell = 1 \dots m$, such that $\bar{\mu}_w = \left( \sum_\ell \lambda_\ell T_\ell^k \right)_\# \mu_k$, where $T_{\ell,\#}^k \mu_k = \mu_\ell$. Hence we have:

$$W_2^2(\bar{\mu}_w, \mu_k) = W_2^2 \left( \left( \sum_\ell \lambda_\ell T_\ell^k \right)_\# \mu_k, \mu_k \right) \leq \sum_{\ell=1}^m \lambda_\ell W_2^2(T_{\ell,\#}^k \mu_k, \mu_k) = \sum_{\ell \neq k} \lambda_\ell W_2^2(\mu_\ell, \mu_k).$$

3) The smoothness energy is a valid interaction energy, by Proposition 7.7 Point 3 (for $W = \|\|\|^2$) (Agueh & Carlier, 2011), it is convex along generalized geodesics and hence by Proposition 7.6 it is barycenter convex:

$$\mathscr{E}(\bar{\mu}_w) \leq \sum_{\ell=1}^m \lambda_\ell \mathscr{E}(\mu_\ell).$$

4) The negative entropy $(-H(\rho))$ is a valid internal energy, by Proposition 7.7 Point 1 ($F(t) = -t \log(t)$) (Agueh & Carlier, 2011), it is convex along generalized geodesics and hence by Proposition 7.6 it is barycenter convex:

$$-H(\bar{\mu}_w) \leq -\sum_{\ell=1}^m \lambda_\ell H(\mu_\ell).$$

$\square$

**Remark 2** (Arithmetic mean and negative entropy). *The negative entropy on discrete space is convex and moreover it is strongly convex with respect to the $\ell^1$ norm hence we have:*

$$-H(\sum_\ell \lambda_\ell \mu_\ell) \leq -\sum_{\ell=1}^m \lambda_\ell H(\mu_\ell) - \frac{1}{2} \sum_{\ell \neq k} \lambda_\ell \lambda_k \|\mu_\ell - \mu_k\|_1^2,$$

*hence the entropy of the arithmetic mean satisfies:*

$$H(\sum_\ell \lambda_\ell \mu_\ell) \geq \sum_{\ell=1}^m \lambda_\ell H(\mu_\ell) + \frac{1}{2} \sum_{\ell \neq k} \lambda_\ell \lambda_k \|\mu_\ell - \mu_k\|_1^2.$$

*hence the diversity of the arithmetic mean depends on the $\ell_1$ distance between the individual models, that does not take in count any semantics.*

**Remark 3.** *Recall strong convexity of $f$ in $\mathbb{R}^d$, there exists $\kappa > 0$:*

$$f(tx + (1-t)y) \leq tf(x) + (1-t)f(y) - (t)(t-1)\kappa \|x - y\|_2^2$$

*To show something about how the entropy of barycenter depends on pairwise distances we need something on "strong geodesic convexity" or "strong barycenter convexity" of negative entropy (Conjectured here):*

$$-H(\bar{\mu}_w) \leq -\sum_{\ell=1}^m \lambda_\ell H(\mu_\ell) - \kappa \sum_{\ell \neq k} \lambda_\ell \lambda_k W_2^2(\mu_\ell, \mu_k)$$

*and equivalently:*

$$H(\bar{\mu}_w) \geq \sum_{\ell=1}^m \lambda_\ell H(\mu_\ell) + \kappa \sum_{\ell \neq k} \lambda_\ell \lambda_k W_2^2(\mu_\ell, \mu_k).$$

*Note that this statement is true for a weaker notion of convexity that is displacement convexity and $\kappa$ depends on Ricci curvature.*

**Lemma 2** (Entropic Regularized W. Barycenters: Controllable entropy via $\varepsilon$). *Assume that $\mu_\ell$ are such that $\mu_{\ell,i} > 0$, for $\ell = 1 \dots m$, $i = 1 \dots M$, we have:*

$$H(\bar{\mu}_w) + H(\mu_\ell) \geq -\sum_{j=1}^N \sum_{i=1}^M \gamma_{ij}^\ell \log(\gamma_{ji}^\ell), \forall \ell = 1 \dots m$$

*Proof.* Let $\gamma \in \mathbb{R}_+^{N \times M}$ be a coupling between $p \in \Delta_M$ and $q \in \Delta_N, q_j > 0$ we have: $\gamma^\top 1_N = p$ and $\gamma 1_M = q$, we have:

$$p_j = \sum_{i=1}^{N} \gamma_{ij} = \sum_{i=1}^{N} q_i \frac{\gamma_{ij}}{q_i}, j = 1 \ldots M$$

Let $a_i = (\frac{\gamma_{ij}}{q_i})_{j=1\ldots M}$. Note that $\sum_{j=1}^{M} \frac{\gamma_{ij}}{q_i} = \frac{q_i}{q_i} = 1$, hence $a_i \in \Delta_M$. Hence $p$ can be written as a convex combination of probabilities:

$$p = \sum_{i=1}^{N} q_i a_i$$

Now the entropy of the convex combination is higher than convex combination of entropies (the entropy is concave):

$$H(p) \geq \sum_{i=1}^{N} q_i H(a_i)$$

$$
\begin{aligned}
H(a_i) &= -\sum_{J=1}^{M} \frac{\gamma_{ij}}{q_i} \log(\frac{\gamma_{ij}}{q_i}) \\
&= -\frac{1}{q_i} \left( \sum_{j=1}^{M} \gamma_{ij} \log(\gamma_{ij}) - \log(q_i)(\sum_{i=1}^{M} \gamma_{ij}) \right) \\
&= -\frac{1}{q_i} \left( \sum_{i=1}^{M} \gamma_{ij} \log(\gamma_{ij}) - \log(q_i) q_i \right).
\end{aligned}
$$

Hence :

$$H(p) \geq -\sum_{j=1}^{N} \sum_{i=1}^{M} \gamma_{ij} \log(\gamma_{ji}) + \sum_{i=1}^{N} q_i \log(q_i) = -\sum_{j=1}^{N} \sum_{i=1}^{M} \gamma_{ij} \log(\gamma_{ij}) - H(q)$$

Hence :

$$H(p) + H(q) \geq -\sum_{i=1}^{N} \sum_{j=1}^{M} \gamma_{ij} \log(\gamma_{ij})$$

$\square$

