# OpenReview forum: "Wasserstein Barycenter Model Ensembling"
_ICLR.cc/2019/Conference_

### Official Review · AnonReviewer1 · 2018-11-02
**review of Wasserstein Barycenter Model Ensembling**

**Rating:** 6
**Confidence:** 4

**Review:**

Paper overview: Model ensembling techniques aim at improving machine learning model prediction results by i) executing several different algorithms on the same task and ii) solving the discrepancies in the responses of all the algorithms, for each task. Some common methods are voting and averaging (arithmetic or geometric average) on the results provided by the different algorithms.
Since averaging amounts to computing barycenters with different distance functions, this paper proposes to use the Wassertein barycenter instead of the L2 barycenter (arithmetic average) or the extended KL barycenter (geometric mean).

Remarks, typos and experiences that would be interesting to add:
     1) Please define the acronyms before using them, for instance DNN (in first page, 4th line), KL (also first page), NLP, etc.
    2) In practice, when ensembling different methods, the geometric and arithmetic mean are not computed with equal weights ($\lambda_l$ in Definition 1). Instead, these weights are computed as the optimal values for a given small dev-set. It would be interesting to see how well does the method compare to these optimal weighted averages, and also if it improves is we also compute the optimal $\lambda_l$ for the Wasserstein barycenter.
    3) How computationally expensive are these methods?
    4) So the output of the ensembling method is a point in the word embedding space, but we know that not all points in this space have an associated word, thus, how are the words chosen?
    5) The image captioning example of Fig.4 is very interesting (although the original image should be added to understand better the different results), can you show also some negative examples? That is to say, when is the Wassertein method is failing but not the other methods.


Points in favor:
     1)Better results: The proposed model is not only theoretically interesting, but it also improves the arithmetic and geometric mean baselines.
    2) Interesting theoretical and practical properties: semantic accuracy, diversity and robustness (see Proposition 1).

Points against: The paper is not easy to read. Ensembling methods are normally applied to the output of a classifier or a regression method, so it is not evident to understand why the 'underlying geometry' is in the word embedding space (page 2 after the Definition 1). I think this is explained in the second paragraph of the paper, but that paragraph is really not clear. I assume that is makes sense to use the word-embedding space for the image caption generation or other ML tasks where the output is a word, but I am not sure how this is used in other cases.

Conclusion: The paper proposes a new method for model assembling by rethinking other popular methods such as the arithmetic and geometric average. It also shows that it improves the current methods. Therefore, I think it presents enough novelties to be accepted in the conference.

---

> ### Author Response · Authors · 2018-11-15
> **Reply to AnonReviewer1**
>
> We thank the reviewer for their positive feedback and their questions that we answer in the following:
>
> 1) REVIEWER:  Please define the acronyms before using them, for instance DNN (in first page, 4th line), KL (also first page), NLP, etc.
>
> AUTHORS: Thanks, we have implemented that in the revision.
>
>   2) REVIEWER: In practice, when ensembling different methods, the geometric and arithmetic mean are not computed with equal weights ($\lambda_l$ in Definition 1). Instead, these weights are computed as the optimal values for a given small dev-set. It would be interesting to see how well does the method compare to these optimal weighted averages, and also if it improves is we also compute the optimal $\lambda_l$ for the Wasserstein barycenter.
>
> AUTHORS: Thanks for the suggestion. We added Appendix B.2 to experiment with this in the multi-label setting, where we set $lambda_l = mAP_{l}/ sum_l mAP_{l}$, i.e lambda_l is proportional to the accuracy of the individual model on the validation set. This is an intuitive approach as one would like to trust more models with higher accuracies on the development set.  This indeed helps all ensembling techniques and maintains the advantage for W. Barycenter on alternative such as arithmetic and geometric as can be seen in Table 11 in Appendix B.2.
>
> 3) REVIEWER: How computationally expensive are these methods?
>
> AUTHORS: Please refer to the reply to AnonReviewer2 above (first point, or to the revised version Appendix E ).
>
> 4) REVIEWER: So the output of the ensembling method is a point in the word embedding space, but we know that not all points in this space have an associated word, thus, how are the words chosen?
>
> AUTHORS: The output of Wasserstein barycenter is still a probability vector (histogram) and not a word embedding. The inputs to W. barycenters are histograms of the models we want to ensemble, we use the geometry of word embeddings to define the Wasserstein distance, the output of the W. ensembling is still an histogram.  The word is chosen as the one with maximal probability for classification and for multi-label prediction we take the top-k (given a varying threshold as defined by average mean precision, or area under the curve).  We use beam search or random beam search for captioning.
>
> 5) REVIEWER: The image captioning example of Fig.4 is very interesting (although the original image should be added to understand better the different results), can you show also some negative examples? That is to say, when is the Wassertein method is failing but not the other methods.
>
>   AUTHORS: We show the image now  thanks for pointing that out.  We have added in Figure 5 (Appendix A), histogram views of how W. Barycenter changes with epsilon for the same example. As epsilon goes to zero we recover geometric mean, as epsilon goes to infinity (entropic regularized ) W.  barycenter becomes close to a uniform distribution, as the probability mass is spread equally across all words. Intermediate values of epsilon allow for a better transfer of mass between semantically related bins, where the radius of neighborhood is defined by epsilon.
>
> 6) REVIEWER: The paper is not easy to read. Ensembling methods are normally applied to the output of a classifier or a regression method, so it is not evident to understand why the 'underlying geometry' is in the word embedding space (page 2 after the Definition 1). I think this is explained in the second paragraph of the paper, but that paragraph is really not clear. I assume that is makes sense to use the word-embedding space for the image caption generation or other ML tasks where the output is a word, but I am not sure how this is used in other cases.
>
> AUTHORS: In the previous version of the paper, we presented an example using word embeddings for building K as a mean to give some intuition about Wasserstein barycenter. Since it may have been confusing, the revised version clarifies that word embeddings are just an example of how to build the cost matrix C (and ultimately K) and that one can define C in different ways depending on the task at hand. For instance, for two semantic classes i, j with respective word embeddings x_i, x_j,  C_{ij}=nor{x_i-x_j}^2 and K_{ij}=e^{-c_{ij}/epsilon}. For other tasks, one may just define K_{ij} as the graph of similarity between semantic classes that can be inferred (for instance) from a knowledge graph, or from wordnet, or from co-occurences, etc. In Section 3 we have now a paragraph "Wasserstein Ensembling in Practice" that discusses Machine learning tasks and their corresponding kernel matrix. Table 12 in the Appendix expands on more tasks and their corresponding K.

---

### Official Review · AnonReviewer2 · 2018-11-04
**a well-written paper with good theoretical and experimental results, but doubts about time-complexity and side information effect**

**Rating:** 6
**Confidence:** 3

**Review:**

The paper proposes a framework based on Wasserstein barycenter to ensemble learning models for a multiclass or a multilabel learning problem. The paper has theoretically shown that the model ensembling using Wasserstein barycenters preserves accuracy, and has a higher entropy than the individual models. Experimental results in the context of attribute-based classification, multilabel learning, and image captioning generation have shown the effectiveness of Wasserstein-based ensembling in comparison to geometric or arithmetic mean ensembling.

The paper is well-written and the experiments demonstrate comparable results. However, the idea of Wasserstein barycenter based ensembling comes at the cost of time complexity since computation of Wasserstein barycenter is more costly than geometric or arithmetic mean. An ensemble is designed to provide lower test error, but also estimate the uncertainty given by the predictions from different models. However, it is not clear how Wasserstein barycenter based ensembling can provide such uncertainty estimate.

Can the authors comment on the time-complexity of the proposed framework in comparison with its baseline methods? Moreover, is it possible to evaluate the uncertainty of predictions with the proposed framework?

In the context of multilabel learning, Frogner et. al. (2015, https://arxiv.org/abs/1506.05439) suggested using Wasserstein distance as a loss function. In the model, they also leverage the side information from word embedding of tag labels. Is the proposed ensembling framework comparable with theirs?

In short, this paper can provide a useful addition to the literature on model ensembling.  Though the proposed framework does improve the performance of predictions in several applications, I am still not fully convinced on time-complexity introduced when computing Wasserstein barycenters.

---

> ### Author Response · Authors · 2018-11-15
> **Reply to  AnonReviewer2**
>
> We thank the reviewer for their positive feedback and address their main concerns:
>
> 1) REVIEWER: Can the authors comment on the time-complexity of the proposed framework in comparison with its baseline methods?
>
> AUTHORS: We added Appendix E to discuss the computational aspects and time complexities. Appendix E.1 discusses the time complexity of a vanilla implementation and how to improve it.
>
> Computing Wasserstein barycenter of histograms of size N, for m models,using Maxiter iterations, needs         O(Maxiter m N^2 ).  Since we have m matrix vector products that costs O(N^2) each. We found Maxiter=5 was enough for convergence. As most of the complexity comes from m, and N, we will then ignore the number of iterations in the discussion.
>
> Alg.1 and Alg.2 are simply parallelizable using m machines which would reduce the cost to O(N^2).
> Note that Kv can be speed up further using low rank approximation of K. K is indeed a kernel matrix that has some low rank structure that can be exploited K =\Phi \Phi^{\top} where Phi \in R^{N\time k}, k<< N (k=polylog(N)), and hence the product can be computed in O(N k) . These improvements have been discussed in a recent paper [1]. Hence, using parallelization and low rank approximation (entropic reg. ) W. barycenter can be computed in near linear time in N.
> Note that using GPU implementation and batching would further improve the overall time complexity as discussed in [2].
>
> Note that we have implemented barycenter in pytorch to enable a GPU computation since most of the time complexity comes from matrix vector product.  Appendix E.2 gives timing experiments on the multi-label task for a vanilla implementation where it is comparedto arithmetic and geometric means (we did not use parallelization, batching nor low rank approximation). We get an average of ~4ms/image for computing the barycenter using a total of m= 8 base models. This is not a big overhead and we are confident that by using the tricks mentioned above (parallelization, batching and low rank approx.) this overhead can be further reduced. However this is beyond the scope of this paper and it is an active area of research on its own.
>
> [1]  J. Altschuler, F. Bach, A. Rudi, and J.Weed. Approximating the Quadratic Transportation Metric in
> Near-Linear Time. ArXiv e-prints , 2018.
> [2] Interpolating between Optimal Transport and MMD using Sinkhorn Divergence.  Feydy et al .
>
>
> 2) REVIEWER: Moreover, is it possible to evaluate the uncertainty of predictions with the proposed framework?
>
> AUTHORS: The uncertainty estimation can be done also with Wasserstein barycenter using bootstrapping. For instance,for image classification we can do K crops and feed them to the m models, and obtain K values for Wasserstein barycenters, then get uncertainty on the Wasserstein barycenter. One other approach is  instead of ensembling deterministic network, one can ensemble stochastic networks (that have noise in their units) and then report means and variances of Wasserstein barycenters.
> Another elegant way of modeling uncertainty in Wasserstein barycenters is by using Gaussian processes or deep Gaussian processes [3, and references therein]. Wasserstein barycenter are well defined between Gaussian processes and is also a Gaussian process [4] .  In summary,one can train individual models that are ( deep ) Gaussian processes and ensemble them using Wasserstein barycenter. The uncertainty modeling  will  carry on to the ensemble since it is also a Gaussian process.
>
> [3] Uncertainty in deep learning. Thesis Yarin Gal.
> [4] Learning from uncertain curves: The 2-Wasserstein metric for Gaussian processes Mallasto et al. https://papers.nips.cc/paper/7149-learning-from-uncertain-curves-the-2-wasserstein-metric-for-gaussian-processes.pdf.
>
> 3) REVIEWER: Relation to Frogner, Zhang et. al
>
> AUTHORS: We have indeed cited Frogner et al. Their work was the first to propose end-to-end learning of deep networks with Wasserstein loss -- nevertheless their algorithm is not an ensembling technique. Their algorithm is a training algorithm, ours is a test time algorithm: Frogner et al. learn a single network  trained with the Wasserstein loss in an end-to-end fashion. Our method allows the ensembling of many models even if they were defined on different output domains (see for example Section 1 where we ensemble attributes to categories)-- this is not achievable in Frogner et al.framework, and is not one of the purposes of their method. Note that our networks are pretrained, and the ensembling is not limited to deep networks as in Frogner et al,  we can be ensembling random forests for instance with Wasserstein barycenters.

---

### Official Review · AnonReviewer3 · 2018-11-04
**averaging label histograms using the W geometry works better than naive (e.g. geometric, arithmetic) averaging.**

**Rating:** 6
**Confidence:** 4

**Review:**

This paper has a simple message. When predicting families (weight vectors) of labels, it makes sense to use an ensemble of predictors and average them using a Wasserstein barycenter, where the ground metric is defined using some a priori knowledge on the labels, here usually distances between word embeddings or more elaborate metrics (or kernels K, as described in p.8). Such barycenters can be easily computed using an algorithm proposed by Benamou et al. 18. When these histograms are not normalized (e.g. their count vectors do not sum to the same quantity) then, as shown by Frogner, Zhang et al, an alternative penalized formulation of OT can be studied, solved numerically with a modified Sinkhorn algorithm, which also leads to a simple W barycenter algorithm as shown by Chizat et al.

The paper starts with a lot of reminders, shows some simple theoretical/stability results on barycenters, underlines the role of the regularization parameter, and then spends a few pages showing that this idea does, indeed, work well to carry out ensemble of multi-tag classifiers.

The paper is very simple from a methodological point of view. Experimental results are convincing, although sometimes poorly presented. Figure are presented in a sloppy way, and a more clear discussion on what K should be used would be welcome, beyond what's proposed in p.8. For these reasons I am positive this result should be published, but I'd expect an additional clarification effort from the authors to reach a publishable draft.

minor comment:
- in remark 1 you mention that as epsilon->0 the solution of Benamou et al. converges to a geometric mean. I would have thought that, on the contrary, the algorithm would have converged to the solution of the true (marginal-regularized) W barycenter. Hence the result your propose is a bit counter-intuitive, could you please develop on that in a future version? Is this valid only because \lambda here is finite? on the contrary, what would happen when eps -> infty then, and K = ones?

- GW for generalized Wasserstein is poor naming. GW usually stands for Gromov-Wasserstein (see Memoli's work).

- \lambda and \lambda_l somewhat clash...

---

> ### Author Response · Authors · 2018-11-15
> **Reply to  AnonReviewer3 part 2/2**
>
>
> 3) REVIEWER: Is this valid only because \lambda here is finite?
>
> AUTHORS : Note that this proof of convergence of the fixed point algorithm to geometric mean holds only for balanced barycenter (Alg. 1) and not for unbalanced ones (Alg. 2) so lambda is not in play here.  Appendix D.2 addresses the case of unbalanced barycenter, where the effect of lambda is more subtle (interpolation between the Hellinger distance and optimal transport (Chizat et al.) ).
>
> 4) REVIEWER: On the contrary, what would happen when eps -> infty then, and K = ones?
>
> AUTHORS : When epsilon goes to infinity in the balanced barycenter case , it is known that Sinkhorn divergence (entropic regularized Wasserstein and normalized with respect to the two measures) converges to MMD distance [2 and citations therein]. K=ones means that all semantic classes become related  (the contrary of epsilon -> 0 where all bins are independent) which will result in diffusive and large spread in the barycenter histogram.  We added Fig 5 in the Appendix A to show a histogram view of how W. Barycenter changes with epsilon (from geometric mean for eps 0, to an almost uniform distribution for eps infinity. Intermediate values allow semantic sharing of mass in the neighborhood defined by eps).
>
> 5) REVIEWER: GW poor naming  --->Thanks.  we changed it  to W_{unb}
> \lambda and \lambda_{ell} clash  --->  while we agree we decided to keep this notation to keep consistent with notations in Chizat et al.

---

> ### Author Response · Authors · 2018-11-15
> **Reply to AnonReviewer3 Part 1/2**
>
> We thank the reviewer for their positive and encouraging feedback and address here their main concerns.
>
> 1) REVIEWER:  Experimental results are convincing, although sometimes poorly presented. Figure are presented in a sloppy way, … discussion on what K should be … beyond what's proposed in p.8.
>
> AUTHORS: We greatly improved in the revised paper, the presentation of experimental results (figures , tables etc). We also clarified how K can be designed for several machine learning tasks that can benefit from W. ensembling. In Section 3, we now have a paragraph "Wasserstein Ensembling in Practice" that discusses this topic. Table 1 discusses multi-class, multi-label and their corresponding K (K is a square matrix, K can be defined through word embeddings or a knowledge graph or confusion/ co-occurrence matrix, a graph constructed based of synonyms or word-net, scene graphs, etc. ) where the histograms are defined on the same domain (source = target).
> We also discuss in Table 1 the "Attributes to Categories" case that unbalanced barycenters allow when source and target domain are different (K rectangular).  As suggested by the reviewer, in order to go beyond the tasks discussed in the paper, we added in the Appendix in Table 12 two additional tasks: Vocabulary expansion   (where base models are NLP models defined on a vocabulary and we would like the ensemble to be defined on a larger vocabulary; K is defined through word embeddings ), Multilingual fusion and translation (base models are NLP models defined on different languages, and the ensemble is defined on yet another language; For this case we need multilingual word embeddings [1] to build K).
>
> [1] Massively multilingual word embeddings. Ammar et al.
>
> 2) REVIEWER:  in remark 1 you mention that "as epsilon->0 the solution of Benamou et al. converges to a geometric mean. I would have thought that, on the contrary, the algorithm would have converged to the solution of the true W Barycenter…. could you please develop on that in a future version?"
>
> AUTHORS: We added Appendix D to discuss this in details. We give now a full proof in Lemma 1 in Appendix D.1  addressing why Algorithm 1 (Benamou et al.) for balanced barycenters converges to geometric mean when epsilon goes to zero. In summary, Algorithm 1 is a fixed point algorithm and when K is identity, it does not recover the non-regularized balanced Wasserstein barycenter since no side information is used and all bins are independent.  When epsilon goes to zero, K approaches identity, and the fixed point algorithm diverges to geometric mean. This does not say that entropic regularization as epsilon goes to zero does not converge to true barycenter -- this was indeed proved in [3] (gamma convergence) without giving an algorithm that achieves this convergence.  We  just say that the fixed point of algorithm (Alg. 1) does not give such a guarantee.
>
> Nevertheless, when epsilon is positive, it is known that the Sinkhorn divergence interpolates between the MMD (maximum mean discrepancy ) distance and Wasserstein distance [2], and hence one can see the barycenter obtained as interpolation between the MMD barycenter and the (marginal reg) true barycenter (as discussed in Appendix D.1).  As we emphasized in the paper, non-zero epsilon is important to control the entropy of the barycenter.
>
> [2] Interpolating between Optimal Transport and MMD using Sinkhorn Divergence.  Feydy et al .
> [3] Convergence of entropic schemes for optimal transport and gradient flows.  Carlier et al.

---

### Author Response · Authors · 2018-11-15
**Revision of the paper uploaded and Comment to Reviewers**

We thank the reviewers for their positive and encouraging feedback. We have uploaded a revised version of the paper to address the questions of the reviewers, where we clarified and added additional experiments at the request of the reviewers:

1) Reviewer 1 and 3: We have clarified that for designing K one is not limited to building it through word embeddings, as explained in the added Table 1 in the paper and in Table 12 in the appendix, for a machine learning task one can design a K given a side information at hand ( with an emphasis that W. Barycenter allows to ensemble histograms defined on different domains.)

2) We added in Appendix B.2 at the request of reviewer 1 an experiment to assess W. Barycenter performance with non uniform weights, that indeed improves accuracy but the advantage for W. Barycenter is maintained.

3) We added Appendix D at the request of reviewer 3 to explain how algorithm 1 behaves as epsilon approaches zero. We provide now a full proof of convergence for alg. 1 of Benamou et al. to geometric mean as K goes to identity.

4) We added Appendix E at the request of reviewer 1 and 2 about the time complexity of W. Barycenter and discussed how to improve it using GPU, batching, parallelization, and low rank approximations (Appendix E.1 ). We also reported timing experiments in multi-label ensembling (Appendix E.2).

We hope those additions and clarifications in the new version address the concerns of the reviewers and improve their overall assessment of the paper.

---

### Meta-Review · Area_Chair1 · 2018-12-14

**Confidence:** 4
**Recommendation:** Accept (Poster)

**Metareview:**

The paper proposes a novel way to ensemble multi-class or multi-label models
based on a Wasserstein barycenter approach. The approach is theoretically
justified and obtains good results. Reviewers were concerned with time
complexity, and authors provided a clear breakdown of the complexity.
Overall, all reviewers were positives in their scores, and I recommend accepting the paper.